# Synergistic Intra- and Cross-Layer Regularization Losses for MoE Expert Specialization

Rizhen Hu [* 1]   Yuan Cao [* 1]   Boao Kong [* 1]   Mou Sun [2]   Kun Yuan [1]

## Abstract

Sparse Mixture-of-Experts (MoE) models scale Transformers efficiently but suffer from expert overlap—redundant representations across experts and routing ambiguity, resulting in severely underutilized model capacity. While architectural solutions like DeepSeekMoE promote specialization, they require substantial structural modifications and rely solely on intra-layer signals. In this paper, we propose two plug-and-play regularization losses that enhance MoE specialization and routing efficiency without modifying router or model architectures. First, an intra-layer specialization loss penalizes cosine similarity between experts' SwiGLU activations on identical tokens, encouraging experts to specialize in complementary knowledge. Second, a cross-layer coupling loss maximizes joint Top-$k$ routing probabilities across adjacent layers, establishing coherent expert pathways through network depth while reinforcing intra-layer expert specialization. Both losses are orthogonal to the standard load-balancing loss and compatible with both the shared-expert architecture in DeepSeekMoE and vanilla top-$k$ MoE architectures. We implement both losses as a drop-in Megatron-LM module. Extensive experiments across pre-training, fine-tuning, and zero-shot benchmarks demonstrate consistent task gains, higher expert specialization, and lower-entropy routing; together, these improvements translate into faster inference via more stable expert pathways. Code is available at https://github.com/xshrz/MoE_loss.

---

[*]Equal contribution   [1]Peking University, Beijing, China [2]Zhejiang Lab, Zhejiang, China. Correspondence to: Mou Sun <123sssmmm@gmail.com>, Kun Yuan <kunyuan@pku.edu.cn>.

*Proceedings of the 43$^{rd}$ International Conference on Machine Learning*, Seoul, South Korea. PMLR 306, 2026. Copyright 2026 by the author(s).

## 1. Introduction

Sparse Mixture-of-Experts (MoE) has emerged as a standard approach for scaling Transformers by expanding parameters while keeping per-token compute roughly constant (Shazeer et al., 2017; Jacobs et al., 1991). In MoE, a learned router activates only a small subset of experts—typically feed-forward networks—for each token (Fedus et al., 2022). From early sparsely gated layers to modern large language models (Du et al., 2022; Fedus et al., 2022; Lepikhin et al., 2020; Zoph et al., 2022; Dai et al., 2024), this design has delivered strong accuracy–efficiency trade-offs. Nevertheless, a fundamental challenge remains: expert specialization progressively deteriorates during training, with tokens routed to different experts exhibiting excessive uniformity and overlap, leading multiple experts to learn redundant knowledge (Dai et al., 2024). This redundancy confronts the router with ambiguous choices among functionally equivalent experts, eroding token-to-expert boundaries and substantially underutilizing model capacity.

Recent work has sought to encourage specialization through architectural modifications. DeepSeekMoE (Dai et al., 2024), HMoE (Wang et al., 2025a), and MoDSE (Sun et al., 2024) redesign expert layouts (e.g., shared or heterogeneous experts) to better match token complexity and balance load, while large-scale routed variants such as Mixtral (Jiang et al., 2024), Mixture of a Million Experts (He, 2024), and ReMoE (Wang et al., 2025b) adjust layer composition, expert granularity, or routing mechanisms to improve accuracy–efficiency trade-offs. These approaches primarily modify architectures and routers and rely on intra-layer dynamics, leaving open whether expert specialization itself can be treated as a first-class training objective.

In contrast to architectural modifications, this paper adopts an orthogonal, loss-centric perspective:

> *Treating expert specialization as a primary training objective rather than a structural property.*

This approach directly shapes expert behavior and complements prior structural innovations. To design these losses, we identify two failure modes of specialization: **(1) Expert Overlap**, where different experts produce nearly identical

activations for the same tokens, creating redundancy. **(2) Routing Ambiguity**, where similar inputs are dispatched inconsistently across experts, indicating ill-defined routing rules. When either occurs, experts collapse toward overlapping knowledge while the router faces ambiguous choices among functionally equivalent experts, undermining the core principle of specialization.

To address these failures, we introduce two complementary regularization loss functions that work in concert:

**(L1) Intra-Layer specialization loss:** This loss penalizes high cosine similarity between different experts' activations for the same token. It discourages functional redundancy and pushes each expert in a layer to develop unique specialization. This targets **Expert Overlap** by discouraging redundant responses from co-activated experts.

**(L2) Cross-Layer coupling loss:** This loss promotes coherent routing across adjacent layers by maximizing the joint probability of top-ranked expert pairs. By encouraging tokens to follow consistent expert sequences through depth, termed expert paths, it sharpens routing distributions, lowers entropy, and enables system-level optimizations such as path-aware placement and caching. This mitigates **Routing Ambiguity** and strengthens depth-wise specialization.

Together, these loss functions translate our diagnosed failure modes into targeted supervision, producing experts that are both functionally distinct within layers and coherently utilized across them.

**Theoretical analysis.** We analyze how the two losses shape expert updates and routing. The intra-layer specialization loss drives co-activated experts toward nearly orthogonal activations and gradients, yielding distinct learning trajectories. The cross-layer coupling loss propagates this specialization across depth under a mild continuity assumption on adjacent-layer representations. Together, the two losses induce a feedback loop in which weak specialization margins sharpen routing, decisive routing purifies each expert's training distribution, and these effects remain compatible with standard load-balancing regularizers.

**Empirical evaluation.** We evaluate specialization and coupling losses across pre-training, supervised fine-tuning, and zero-shot evaluation for both vanilla and DeepSeek-style MoE architectures at multiple scales. On pre-training, adding sp + cp on top of load-balancing consistently reduces perplexity, with up to $\sim 2.7\%$ relative improvement. In scaling experiments, activating only $N{=}6$ experts already outperforms the baseline even with $N{=}10$. The gains transfer to downstream adaptation: under LoRA SFT on $\sim 16$B-class MoE backbones, our method improves average downstream performance by $+3.9$ points on DeepSeek-MoE and $+4.6$ points on DeepSeek-V2-Lite (averaged over eight benchmarks), and in full-parameter fine-tuning of

Qwen3-30B-A3B we further boost HumanEval pass@1 by $+3.66$ points. Across settings, improvements correlate with stronger structural signals—lower activation overlap, lower routing entropy, and clearer cross-layer expert paths—and persist across random seeds and hyperparameter choices.

**Core contributions.** Our contributions are listed as follows:

**(C1) Loss-centric specialization.** We address expert overlap and routing ambiguity with two complementary regularization losses: an intra-layer term penalizing same-token activation similarity and a cross-layer term encouraging coherent expert paths through depth.

**(C2) Theory for specialization and coupling.** We theoretically show that the intra-layer specialization loss drives co-activated experts toward orthogonality and that cross-layer coupling propagates specialization across depth while remaining compatible with load-balancing objectives.

**(C3) Accuracy, specialization, and efficiency gains.** Across pre-training, fine-tuning, and zero-shot experimental evaluations, the two losses consistently improve perplexity, downstream accuracy, expert-specialization metrics, and inference throughput under expert parallelism.

**(C4) Plug-and-play integration.** Both losses are router- and architecture-agnostic, implemented as a drop-in Megatron-LM module that requires only a configuration flag and no changes to core model code. This design enables seamless integration into existing MoE training pipelines.

## 2. Related Works

Here we discuss prior works which are closely related to our proposed approach. More related works can be found in Appendix A.

**Balancing losses and specialization objectives.** A primary strategy to prevent routing collapse and improve stability in MoE training is to enforce balanced expert utilization. Early systems such as GShard (Lepikhin et al., 2020) and Switch (Fedus et al., 2022) introduced auxiliary load-balancing terms to distribute tokens across experts, with router z-loss (Zoph et al., 2022) providing additional stabilization. BASE layers (Lewis et al., 2021) formulated routing as an optimal linear assignment problem, achieving perfectly balanced usage without auxiliary terms. Expert-Choice routing (Zhou et al., 2022) further reversed the assignment process, allowing experts to select their Top-$k$ tokens, which inherently balances load. These methods primarily regulate *how much* each expert is used. In contrast, our approach is complementary: we supervise *what* experts learn and *how* their paths align, introducing a within-layer similarity penalty to discourage activation overlap and a cross-layer coupling term to enforce coherence, while leaving existing balancing mechanisms intact.

**Architectural and router-Level approaches.** Another line of work promotes expert specialization by redesigning MoE architectures or router mechanisms. DeepSeekMoE (Dai et al., 2024) partitions experts more finely and introduces always-active shared experts, allowing routed specialists to focus on idiosyncratic patterns. Router-centric methods also refine gating: ReMoE (Wang et al., 2025b) replaces Top-$k$ Softmax with a differentiable ReLU router and adaptive $L_1$ regularization, while Dynamic MoE (Guo et al., 2025b) auto-tunes both the number of activated experts per token and the size of the expert pool. Several structural variants further expand capacity and specialization. Mixtral layers multiple FFNs with top-2 routing, achieving strong accuracy–efficiency trade-offs (Jiang et al., 2024); Mixture of a Million Experts pushes expert granularity to the extreme (He, 2024); HMoE mixes experts of different sizes and biases usage toward smaller ones to encourage division of labor (Wang et al., 2025a); MoDSE deploys diverse-sized experts with pairwise allocation to stabilize routing and balance compute across devices (Sun et al., 2024); while simpler approaches such as Hash Layers (Roller et al., 2021) and THOR (Zuo et al., 2021) enforce balanced usage through fixed or randomized routing schemes. Unlike these methods—which modify layer composition or router design and largely rely on in-layer dynamics—our approach is architecture-agnostic. We impose explicit specialization objectives, namely a within-layer similarity penalty and a cross-layer coupling loss, on top of existing designs without altering attention, FFN, or router code paths.

## 3. Mixture-of-Experts Models: Preliminaries

**MoE layer.** An MoE layer replaces the dense feed-forward network (FFN) in a Transformer block with $E$ experts and a router. For the $i$-th token at layer $\ell$, the computation proceeds in three steps.

**Step 1. Routing.** Let $x_i^{(\ell)} \in \mathbb{R}^h$ denote the input representation. The router computes a logit $q_i^{(\ell,e)}$ for each expert $e$ using a learnable routing vector $r^{(\ell,e)} \in \mathbb{R}^h$, then applies softmax to obtain routing scores

$$q_i^{(\ell,e)} = \langle x_i^{(\ell)}, r^{(\ell,e)}\rangle, \quad s_i^{(\ell,e)} := \frac{\exp(q_i^{(\ell,e)})}{\sum_{j=1}^{E}\exp(q_i^{(\ell,j)})}, \quad (1)$$

where $\langle \cdot, \cdot \rangle$ denotes the inner product.

**Step 2. Expert processing.** The router activates the top-$k$ experts $A_i^{(\ell)} \subseteq \{1, \dots, E\}$ for token $x_i^{(\ell)}$. Each expert is an FFN, typically implemented as SwiGLU, with parameters $(W_{\text{gate}}^{(\ell,e)}, W_{\text{up}}^{(\ell,e)}, W_{\text{down}}^{(\ell,e)})$. The expert computation is:

$$\begin{aligned} z_i^{(\ell,e)} &= \text{Swish}\left(W_{\text{gate}}^{(\ell,e)} x_i^{(\ell)}\right) \odot \left(W_{\text{up}}^{(\ell,e)} x_i^{(\ell)}\right), \\ y_i^{(\ell,e)} &= W_{\text{down}}^{(\ell,e)} z_i^{(\ell,e)}, \end{aligned} \quad (2)$$

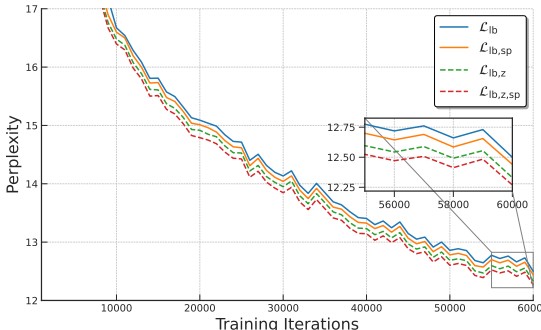

*Figure 1.* The perplexity for training a 1.1B model with different regularization. Setup is in Table 4.

where $z_i^{(\ell,e)}$ denotes the intermediate activation, $y_i^{(\ell,e)}$ is the expert output, and $\odot$ denotes element-wise multiplication.

**Step 3. Combination.** The layer output is a weighted sum of activated experts:

$$y_i^{(\ell)} = \sum_{e \in \mathbb{A}_i^{(\ell)}} s_i^{(\ell,e)} y_i^{(\ell,e)}. \quad (3)$$

**Properties of well-behaved MoE.** Putting these pieces together, we use "well-behaved MoE" as convenient shorthand for layers that approximately satisfy two qualitative properties: (i) experts are functionally disjoint on the token distribution, and (ii) routing is decisive and low-entropy. These properties are not formal requirements but rather capture the behaviors our losses are designed to encourage.

When these properties break down, we observe the two empirical failure modes introduced in Section 1. Expert overlap arises when experts in $A_i^{(\ell)}$ produce highly similar activations $z_i^{(\ell,e)}$ on the same token, rendering their contributions nearly mergeable and wasting model capacity. Routing ambiguity occurs when near-identical inputs $x_i^{(\ell)}$ are dispatched to different experts, blurring token–expert boundaries and preventing experts from developing distinct roles. When either failure mode occurs, experts collapse toward redundant representations while the router faces ambiguous choices among functionally equivalent experts. The intra-layer specialization loss (Section 4) and the cross-layer coupling loss (Section 5) are designed to directly counteract these two pathologies.

## 4. Intra-layer specialization loss

This section introduces an intra-layer regularization loss that targets *expert functional overlap* within each MoE layer. Standard load-balancing losses enforce even utilization, but do not prevent multiple experts from learning redundant transformations on the *same* tokens, which weakens the intended division of labor in MoE.

**Functional view of expert overlap.** Following the view of FFN blocks as concept extractors and writers to the residual stream (Geva et al., 2022), we decompose each expert into an intermediate activation $z_i^{(\ell,e)}$ and a down-projection $W_{\text{down}}^{(\ell,e)}$ as in Eq. (2). If two co-activated experts produce nearly identical intermediates $z_i^{(\ell,e)} \approx z_i^{(\ell,\nu)}$ on the same token, their contributions become algebraically mergeable on that token and the experts are functionally redundant.

**Loss definition.** Motivated by this view, we discourage same-token similarity in the intermediate activations. For token $x_i$ we define the intra-layer specialization loss

$$\mathcal{R}_{\text{sp}}(x_i) = \sum_{\ell=1}^{L} \sum_{e \neq \nu \in \mathbb{A}_i^{(l)}} \left[ \cos\left(z_i^{(\ell,e)}, z_i^{(\ell,\nu)}\right) \right]^2. \quad (4)$$

where $\mathbb{A}_i^{(\ell)}$ is the set of experts activated for $x_i$ at layer $\ell$. Squaring the cosine emphasizes highly overlapping pairs while keeping the penalty smooth. Crucially, the specialization loss $\mathcal{R}_{\text{sp}}$ only acts on experts that are *co-activated on the same token and layer*, leaving shared features across different tokens or contexts unconstrained.

**Effect on expert updates.** Beyond interpretability, the intermediate activation $z_i^{(\ell,e)}$ also governs how expert $e$ is updated through its down-projection matrix $W_{\text{down}}^{(\ell,e)}$. The next proposition makes this connection explicit.

**Proposition 4.1** (Activation-gradient alignment). *For any two activated experts $e, \nu \in \mathbb{A}_i^{(\ell)}$, the cosine similarity between the gradients of the total loss $\mathcal{L}$ with respect to their down-projection matrices satisfies*

$$\cos\left( \frac{\partial \mathcal{L}}{\partial W_{\text{down}}^{(\ell,e)}}, \frac{\partial \mathcal{L}}{\partial W_{\text{down}}^{(\ell,\nu)}} \right) = \cos\left( z_i^{(\ell,e)}, z_i^{(\ell,\nu)} \right), \quad (5)$$

*where $z_i^{(\ell,e)}$ and $z_i^{(\ell,\nu)}$ denote the corresponding intermediate activations. (The proof is provided in Appendix B.1.)*

Proposition 4.1 links representation geometry to optimization dynamics: for co-activated experts on the same token, the cosine similarity of their activations $z_i^{(\ell,\cdot)}$ *exactly* equals the cosine similarity of their $W_{\text{down}}$ gradients. We summarize the key implication for expert updates as follows:

> *Penalizing same-token activation similarity makes co-activated experts' $W_{\text{down}}$ gradients more orthogonal, driving distinct learning trajectories and strengthening intra-layer functional specialization.*

**Empirical impact.** To validate that $\mathcal{R}_{\text{sp}}$ captures specialization and improves training, we pre-train a 1.1B MoE model (100M activated parameters) with and without this regularizer while keeping all other settings identical. We

denote by $\mathcal{L}_{(\cdot)}$ the full training objective, i.e., the language-modeling loss together with the indicated regularization terms. We evaluate four configurations: $\mathcal{L}_{\text{lb}}$ (load balancing only), $\mathcal{L}_{\text{lb,sp}}$ (load balancing + specialization), $\mathcal{L}_{\text{lb,z}}$ (load balancing + $z$-loss (Zoph et al., 2022)), and $\mathcal{L}_{\text{lb,z,sp}}$ (all three losses involved). Figure 1 shows that incorporating $\mathcal{R}_{\text{sp}}$ consistently reduces perplexity, with the combined $\mathcal{L}_{\text{lb,z,sp}}$ achieving the best performance.

# 5. Cross-layer coupling loss

This section introduces a cross-layer coupling loss to address routing ambiguity and propagate expert specialization across depth. When near-identical tokens are dispatched to different experts, each expert receives a mixed and largely overlapping data distribution; consequently, their gradients become correlated and updates drive them toward similar functionality. Without stable, consistent assignments, experts cannot develop distinct roles, token–expert boundaries remain blurred, and the intended division of labor in MoE collapses into redundant behavior.

**Cross-layer coupling in pre-trained MoE models.** Recent work has identified an emergent property in MoE known as *cross-layer coupling* (Cai et al., 2024; Yao et al., 2024): routing decisions in adjacent layers are strongly correlated, such that the expert activated at layer $\ell$ is highly predictive of the expert activated at layer $\ell+1$. During training, models spontaneously develop such structured pathways, forming coherent information pipelines through depth.

**Cross-layer coupling as a specialization amplifier.** Cross-layer coupling clearly promotes routing stability: when tokens consistently traverse fixed expert sequences (e.g., "expert 3 in layer 7 followed by expert 5 in layer 8"), routing ambiguity is eliminated by definition. However, its effect on expert specialization is less obvious. Specifically, we ask: *how does inter-layer structural consistency influence intra-layer expert differentiation?* In other words, if tokens follow stable paths across layers, does this help individual experts within each layer become more specialized? Our analysis reveals that the answer is yes, through a propagation mechanism formalized below.

**Proposition 5.1.** *Let $\mathbb{A}_i^{(\ell)}$ denote the set of activated experts for token $x_i$ at layer $l$. Consider two adjacent layers $\ell$ and $\ell + 1$ that satisfy the following conditions:*

*1. **Representation continuity.** For a token $x_i$, its representations evolve smoothly across layers: $\cos(x_i^{(\ell)}, x_i^{(\ell+1)}) \geq 1 - \delta^2$ for small $\delta \in (0, 1)$.*

*2. **Source-layer specialization.** Layer $\ell$ exhibits expert specialization with nearly orthogonal router weights: for experts $e_1 \in \mathbb{A}_i^{(\ell)}$ and $e_2 \in \mathbb{A}_j^{(\ell)}$ processing different tokens $x_i \neq x_j$, we have $|\cos(r^{(\ell,e_1)}, r^{(\ell,e_2)})| \leq \varepsilon$ for a small*

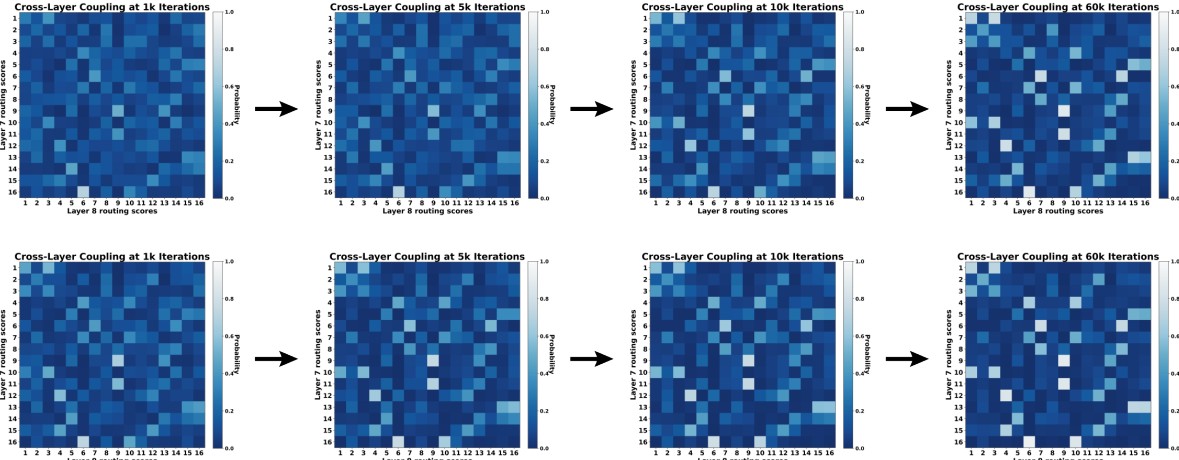

*Figure 2.* Conditional activation probabilities between experts in layers 7 and 8 for a 0.4B MoE model. *Top:* training with only load-balance regularization. *Bottom:* training with both load-balance and coupling regularization.

$\varepsilon \in (0, 1)$.

**3. Strong cross-layer coupling.** *Adjacent layers exhibit stable expert pathways with high routing correlation. For any expert $e \in \mathbb{A}_i^{(\ell)}$ activated by token $x_i$, there exists a corresponding expert $\nu \in \mathbb{A}_i^{(\ell+1)}$ such that both routing decisions are confident: $\cos(x_i^{(\ell)}, r^{(\ell,e)}) \geq 1 - \iota^2$ and $\cos(x_i^{(\ell+1)}, r^{(\ell+1,\nu)}) \geq 1 - \iota^2$ for small $\iota \in (0, 1)$.*

*Under these conditions, layer $\ell + 1$ inherits the specialization structure from layer $\ell$:*

$$\left| \cos\left( r^{(\ell+1,\nu_1)}, r^{(\ell+1,\nu_2)} \right) \right| \leq \varepsilon + O(\delta, \iota) \qquad (6)$$

*for experts $\nu_1 \in \mathbb{A}_i^{(\ell+1)}$ and $\nu_2 \in \mathbb{A}_j^{(\ell+1)}$ processing different tokens, where the error term $O(\delta, \iota)$ vanishes as $\delta$ and $\iota$ decrease to 0 (proof in Appendix B.2).*

Proposition 5.1 shows that when layer $\ell$ has well-specialized experts (Condition 2) and is strongly coupled to layer $\ell + 1$ (Condition 3), the specialization structure is transferred to the adjacent layer with bounded degradation (Eq. (6)). Localized specialization can therefore cascade through depth, yielding globally specialized representations. This motivates the following summary.

> *Cross-layer coupling acts as a **specialization amplifier**: it transforms localized expert differentiation into network-wide functional diversity by creating stable pathways that propagate specialization across depth.*

**Cross-layer coupling loss.** Although cross-layer coupling emerges naturally, it develops slowly and incompletely, especially early in training when routing ambiguity is severe. Rather than waiting for it to appear organically, we introduce a coupling regularizer $\mathcal{R}_{\text{cp}}$ that directly maximizes joint rout-

ing probabilities between adjacent layers. For each token $x_i$, we define the pathway strength between expert $e$ in layer $\ell$ and expert $\nu$ in layer $\ell + 1$ as $P_i^{(\ell,(e,\nu))} = s_i^{(\ell,e)} s_i^{(\ell+1,\nu)}$, the product of their routing scores. The loss focuses on the Top-$k$ strongest cross-layer connections for each activated expert:

$$\mathcal{R}_{\text{cp}}(x_i) = -\sum_{\ell=1}^{L-1} \sum_{e=1}^{E} \sum_{\nu \in \mathbb{T}_i^{(\ell,e)}} P_i^{(\ell,(e,\nu))},$$

$$\text{where} \quad P_i^{(\ell,(e,\nu))} = s_i^{(\ell,e)} s_i^{(\ell+1,\nu)}. \qquad (7)$$

Here $s_i^{(\ell,e)}$ is defined in Eq. (1), and $\mathbb{T}_i^{(\ell,e)}$ contains the $k$ experts in layer $\ell + 1$ with the highest joint probabilities with expert $e$. Minimizing $\mathcal{R}_{\text{cp}}$ encourages decisive, high-probability pathways between adjacent layers, and by Proposition 5.1 these pathways create the structural conditions for specialization to propagate throughout the network.

By concentrating probability mass on a few strong cross-layer expert pairs, $\mathcal{R}_{\text{cp}}$ reduces routing ambiguity and encourages consistent expert identities across depth. This lowers token-distribution overlap among experts, decreases gradient sharing, and promotes divergent specialization, while remaining complementary to standard load-balancing objectives discussed in Section 6.

**Empirical validation.** To verify that cross-layer coupling is both natural and worth amplifying, we pre-train a 0.4B MoE model with 80M activated parameters and monitor conditional activation probabilities between adjacent layers. As shown in Figure 2, a clear coupling structure is already present early in training and becomes more pronounced over time, confirming that structured expert paths are an intrinsic feature of MoE learning.

# 6. Unified Training Objective and Theoretical Picture

**Joint training objective.** With the intra-layer specialization loss $\mathcal{R}_{\text{sp}}$ (Section 4) and the cross-layer coupling loss $\mathcal{R}_{\text{cp}}$ (Section 5), we train MoE models by adding them as plug-in regularizers on top of the standard MoE objective. For each token $x_i$, the full training objective is

$$\mathcal{L}_{\text{lb,sp,cp}}(x_i) := \underbrace{\mathcal{L}(x_i) + \mathcal{R}_{\text{lb}}(x_i)}_{\text{Load balance loss}} \\ + \underbrace{\lambda_{\text{sp}}\,\mathcal{R}_{\text{sp}}(x_i) + \lambda_{\text{cp}}\,\mathcal{R}_{\text{cp}}(x_i)}_{\text{Intra-/Cross-layer regularization}}, \quad (8)$$

where $\mathcal{L}(x_i)$ is the primary language modeling loss, $\mathcal{R}_{\text{lb}}(x_i)$ is the standard load-balancing regularizer, and $\lambda_{\text{sp}}, \lambda_{\text{cp}}$ control the strength of specialization and coupling regularization. We use $\mathcal{L}_{(\cdot)}$ to denote the full objective with the indicated regularizers (e.g., $\mathcal{L}_{\text{lb}}$, $\mathcal{L}_{\text{lb,sp}}$, $\mathcal{L}_{\text{lb,cp}}$, and $\mathcal{L}_{\text{lb,sp,cp}}$). In practice, router-stabilization terms such as $z$-loss can be included on top of Eq. (8); our two losses only reuse intermediate activations and routing scores already produced in the forward pass and therefore require no architectural or routing-code modifications.

**Practical overhead.** Both $\mathcal{R}_{\text{sp}}$ and $\mathcal{R}_{\text{cp}}$ are lightweight auxiliaries that operate on quantities already produced in standard MoE training. $\mathcal{R}_{\text{sp}}$ only uses the Top-$k$ activated experts per token and adds an $O(k^2 d)$ similarity computation, which can reuse cached expert activations from the forward pass. $\mathcal{R}_{\text{cp}}$ is computed from scalar routing scores and introduces no additional matrix multiplications. A detailed compute/memory accounting and a empirical validation are all provided in Appendix H.

**Core theoretical picture: why the two losses form a coherent system.** Sections 4 and 5 introduced $\mathcal{R}_{\text{sp}}$ and $\mathcal{R}_{\text{cp}}$ as direct supervision signals for expert overlap and routing ambiguity. At a high level, Proposition 4.1 shows that reducing same-token activation similarity directly decorrelates co-activated experts' update directions through $W_{\text{down}}$, while Proposition 5.1 shows that strong cross-layer coupling can propagate specialization structure across depth with bounded degradation. Moreover, our regularizers remain compatible with standard load balancing (formal constructions in Appendix C). We also empirically verify that adding $\mathcal{R}_{\text{sp}}$ and $\mathcal{R}_{\text{cp}}$ does not destabilize load balancing by analyzing load-balance loss curves throughout training (Appendix D.4).

**Closed-loop theory: specialization and routing sharpen each other.** Beyond the local effects above, our key new theoretical contribution is a closed-loop mechanism linking specialization and routing. In this loop, even a mild best-expert loss advantage tends to make routing more decisive

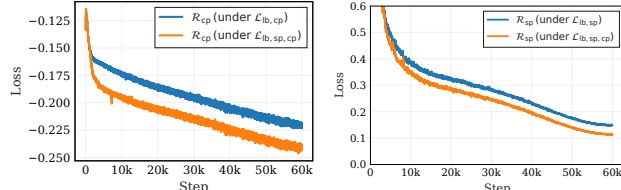

*Figure 3.* Training dynamics on the 0.4B MoE model. Left: cross-layer coupling loss $\mathcal{R}_{\text{cp}}$ when training with $\mathcal{L}_{\text{lb,cp}}$ vs. $\mathcal{L}_{\text{lb,cp,sp}}$; adding $\mathcal{R}_{\text{sp}}$ consistently makes $\mathcal{R}_{\text{cp}}$ more negative (stronger coupling). Right: intra-layer specialization loss $\mathcal{R}_{\text{sp}}$ when training with $\mathcal{L}_{\text{lb,sp}}$ vs. $\mathcal{L}_{\text{lb,sp,cp}}$; adding $\mathcal{R}_{\text{cp}}$ consistently reduces $\mathcal{R}_{\text{sp}}$ (stronger specialization).

(lower entropy), and more decisive routing in turn purifies each expert's effective training data, which strengthens regional experts and further amplifies specialization. Formally, Theorem C.1 establishes the first direction (weak advantage sharpens routing), while Theorem C.4 establishes the reverse direction (sharp and stable routing amplifies specialization on advantage regions). This perspective also clarifies why our two losses are complementary: $\mathcal{R}_{\text{sp}}$ helps create and maintain meaningful expert advantages by discouraging within-layer overlap, while $\mathcal{R}_{\text{cp}}$ stabilizes token–expert paths across depth so that purity and specialization persist and propagate.

**A quick empirical check: $\mathcal{R}_{\text{sp}}$ and $\mathcal{R}_{\text{cp}}$ reinforce each other.** The closed-loop picture above suggests that specialization and coupling should interact positively rather than compete. We verify this interaction in a lightweight experiment on the same 0.4B MoE setting used in Section 5. We first train the model with $\mathcal{L}_{\text{lb,cp}}$ and $\mathcal{L}_{\text{lb,cp,sp}}$, respectively. The coupling loss values in Figure 3 show that adding $\mathcal{R}_{\text{sp}}$ consistently lowers $\mathcal{R}_{\text{cp}}$, indicating that stronger intra-layer specialization promotes clearer cross-layer coupling. Conversely, we train the model with $\mathcal{L}_{\text{lb,sp}}$ and $\mathcal{L}_{\text{lb,sp,cp}}$, respectively. The specialization loss values in Figure 3 show that adding $\mathcal{R}_{\text{cp}}$ consistently lowers $\mathcal{R}_{\text{sp}}$, indicating that stabilizing cross-layer pathways reinforces within-layer expert differentiation.

**Takeaway: a positive feedback loop.** Combining Theorem C.1 (weak specialization pushes routing to be decisive) and Theorem C.4 (decisive routing amplifies specialization), we obtain a positive feedback loop: even a weak loss advantage of the best expert tends to sharpen routing (lower entropy), which increases the purity of each expert's effective training data; this purer training distribution then strengthens regional experts and further amplifies specialization. This loop complements our objectives in Sections 4 and 5: $\mathcal{R}_{\text{sp}}$ directly encourages functional differentiation within a layer, while $\mathcal{R}_{\text{cp}}$ stabilizes pathways across depth so that these specialization signals persist and propagate. Figure 4 provides a schematic summary of this mechanism.

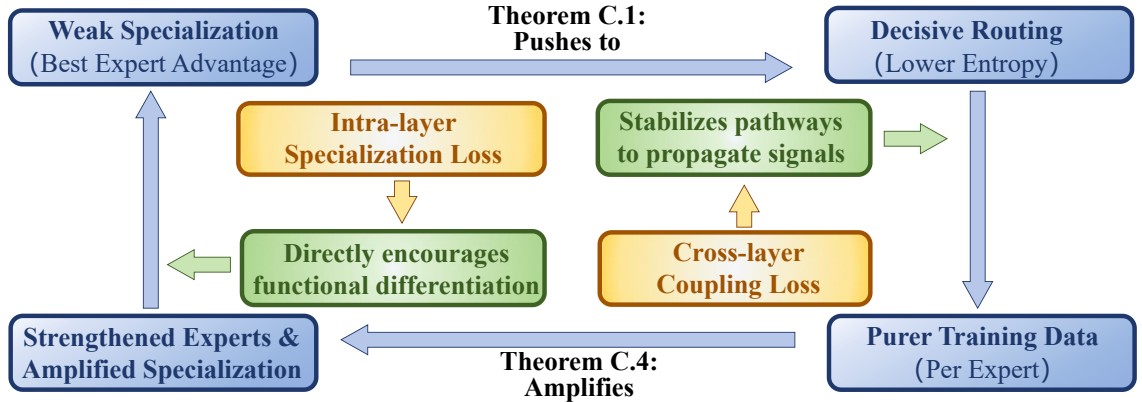

*Figure 4.* **The self-reinforcing cycle** illustrating how expert specialization and routing decisiveness amplify one another.

*Table 1.* Validation perplexity ($\downarrow$) across model scales and auxiliary-loss configurations.

| | Vanilla MoE | | | DeepSeek-style MoE | | |
|---|---|---|---|---|---|---|
| **Losses** | **Small** | **Medium** | **Large** | **Small** | **Medium** | **Large** |
| $\mathcal{L}_{\mathrm{lb}}$ | 14.01 | 12.50 | 9.68 | 13.54 | 12.33 | 9.56 |
| $\mathcal{L}_{\mathrm{lb,o,v}}$ | 14.27 | 12.71 | 9.84 | 13.76 | 12.51 | 9.81 |
| $\mathcal{L}_{\mathrm{lb,sp,cp}}$ | 13.75 | 12.27 | 9.48 | 13.37 | 12.16 | 9.47 |
| $\mathcal{L}_{\mathrm{lb,z}}$ | 13.80 | 12.33 | 9.52 | 13.40 | 12.07 | 9.46 |
| $\mathcal{L}_{\mathrm{lb,z,o,v}}$ | 14.15 | 12.67 | 9.82 | 13.69 | 12.36 | 9.77 |
| $\mathcal{L}_{\mathrm{lb,z,sp,cp}}$ | 13.63 | 12.17 | 9.42 | 13.30 | 11.99 | 9.39 |

# 7. Experiments

In this section, we present a series of experiments to validate the performance of the proposed approaches. Additional details and additional experimental results can be found in Appendix D.

**Comparison of validation perplexity.** We evaluate on the C4 dataset (Raffel et al., 2020) across three model scales for both Vanilla MoE and DeepSeek-style MoE. Table 1 summarizes validation perplexity under different auxiliary-loss configurations. Beyond standard load-balancing (and optionally the router $z$-loss), we include a recent auxiliary-loss baseline from Guo et al. (2025a) that regularizes orthogonality and routing-logit variance (denoted as $\mathcal{L}_{\mathrm{lb,o,v}}$, and $\mathcal{L}_{\mathrm{lb,z,o,v}}$ when combined with $z$-loss); further discussions for auxiliary-loss comparisons are deferred to Appendix G. Overall, our specialization-and-coupling regularizers consistently improve perplexity across scales and architectures, while the variance-on-logits baseline tends to degrade perplexity relative to the corresponding objectives. These trends hold both with and without $z$-loss, indicating that our activation/path-based regularization is complementary to standard router-stabilization and more robust than directly amplifying routing-logit dispersion. Notably, the gains are architecture-agnostic: the same objectives benefit both Vanilla MoE and DeepSeek-style MoE (with

shared experts), and in some regimes the improved Vanilla MoE reaches or surpasses the DeepSeek-style baseline without increasing activated capacity, highlighting that targeted training objectives can rival architectural router modifications while remaining plug-and-play. We further report controlled ablations (Appendix D.2), three-seed repetitions (Appendix D.5), and hyperparameter sensitivity sweeps (Appendix D.6) to verify that these trends are robust beyond the main settings. After pre-training, we additionally benchmark the resulting checkpoints in a controlled zero-shot evaluation suite to measure downstream generalization; details and full results are provided in Appendix D.3.

**LoRA SFT Downstream Evaluation (16B-class MoE Backbones).** We assess downstream performance under a LoRA SFT setting on three ∼16B-scale MoE backbones— DeepSeek-MoE-16B (Dai et al., 2024), DeepSeek-V2-Lite (Liu et al., 2024a), and Ling-mini-2.0 (Tian et al., 2025)—evaluated on MMLU, MMLU-Pro, HellaSwag, BBH, GPQA-Diamond, MBPP, HumanEval, and GSM8K (Table 2). To ensure a fair comparison, we match each backbone's *auxiliary routing regularization* to its pre-training recipe. Specifically, DeepSeek backbones use load-balancing as the baseline auxiliary loss. Ling uses Auxiliary-Loss-Free load balance as the baseline, and we evaluate AuxLossFree+cp+sp. We also include the variance-on-logits auxiliary baseline of Guo et al. (2025a), which adds variance-related regularizers on top of the same $\mathcal{L}_{\mathrm{CE}}$ (and on top of $\mathcal{L}_{\mathrm{lb}}$ for DeepSeek), with standard stabilization to complete fine-tuning.

Across all three backbones, adding our coupling and specialization regularizers delivers the strongest overall results: for both DeepSeek-MoE and DeepSeek-V2-Lite, it consistently improves broad knowledge and hard reasoning benchmarks and yields non-trivial gains on code generation and arithmetic reasoning compared to the matched baseline, while also surpassing the variance-on-logits method on most metrics. Under the AuxLossFree-controlled setting, Ling-mini-2.0 exhibits the same qualitative pattern, where adding

*Table 2.* Downstream evaluation results (mean ± stderr) across multiple models.

| Method | Model | MMLU | MMLU-Pro | HellaSwag | BBH | GPQA-Diamond | MBPP | HumanEval | GSM8K |
|---|---|---|---|---|---|---|---|---|---|
| $\mathcal{L}_{lb}$ | | $0.4143_{\pm 0.0040}$ | $0.1729_{\pm 0.0034}$ | $0.5852_{\pm 0.0049}$ | $0.4041_{\pm 0.0053}$ | $0.2323_{\pm 0.0301}$ | $\mathbf{0.4180}_{\pm 0.0221}$ | $0.2927_{\pm 0.0356}$ | $0.2661_{\pm 0.0122}$ |
| $\mathcal{L}_{lb,z}$ | DeepSeek-MOE | $0.3717_{\pm 0.0040}$ | $0.1410_{\pm 0.0032}$ | $0.5621_{\pm 0.0050}$ | $0.3864_{\pm 0.0053}$ | $0.2273_{\pm 0.0299}$ | $0.3200_{\pm 0.0209}$ | $0.2500_{\pm 0.0339}$ | $0.1789_{\pm 0.0106}$ |
| $\mathcal{L}_{lb,o,v}$ | | $0.4293_{\pm 0.0041}$ | $0.1735_{\pm 0.0034}$ | $0.5882_{\pm 0.0049}$ | $0.4386_{\pm 0.0055}$ | $0.2626_{\pm 0.0314}$ | $0.3940_{\pm 0.0219}$ | $0.2805_{\pm 0.0352}$ | $0.2896_{\pm 0.0125}$ |
| $\mathbf{\mathcal{L}_{lb,sp,cp}}$ | | $\mathbf{0.4586}_{\pm 0.0041}$ | $\mathbf{0.2276}_{\pm 0.0034}$ | $\mathbf{0.5906}_{\pm 0.0049}$ | $\mathbf{0.4558}_{\pm 0.0053}$ | $\mathbf{0.2828}_{\pm 0.0321}$ | $0.4100_{\pm 0.0220}$ | $\mathbf{0.3415}_{\pm 0.0371}$ | $\mathbf{0.3275}_{\pm 0.0128}$ |
| $\mathcal{L}_{lb}$ | | $0.5361_{\pm 0.0040}$ | $0.2605_{\pm 0.0039}$ | $0.5893_{\pm 0.0049}$ | $0.4346_{\pm 0.0056}$ | $0.2879_{\pm 0.0323}$ | $0.3680_{\pm 0.0216}$ | $0.3110_{\pm 0.0363}$ | $0.4617_{\pm 0.0137}$ |
| $\mathcal{L}_{lb,z}$ | DeepSeek-V2-Lite | $0.5268_{\pm 0.0040}$ | $0.2103_{\pm 0.0037}$ | $0.5777_{\pm 0.0049}$ | $0.3860_{\pm 0.0054}$ | $0.2778_{\pm 0.0319}$ | $0.3660_{\pm 0.0216}$ | $0.3049_{\pm 0.0363}$ | $0.4147_{\pm 0.0136}$ |
| $\mathcal{L}_{lb,o,v}$ | | $0.5474_{\pm 0.0040}$ | $0.2523_{\pm 0.0039}$ | $0.5902_{\pm 0.0049}$ | $0.4264_{\pm 0.0056}$ | $0.3131_{\pm 0.0330}$ | $0.4080_{\pm 0.0216}$ | $0.3049_{\pm 0.0361}$ | $0.4701_{\pm 0.0137}$ |
| $\mathbf{\mathcal{L}_{lb,sp,cp}}$ | | $\mathbf{0.5735}_{\pm 0.0040}$ | $\mathbf{0.3108}_{\pm 0.0039}$ | $\mathbf{0.6091}_{\pm 0.0049}$ | $\mathbf{0.4793}_{\pm 0.0055}$ | $\mathbf{0.3535}_{\pm 0.0341}$ | $\mathbf{0.4280}_{\pm 0.0216}$ | $\mathbf{0.3598}_{\pm 0.0371}$ | $\mathbf{0.5004}_{\pm 0.0138}$ |
| AuxLossFree | | $0.6998_{\pm 0.0036}$ | $0.4759_{\pm 0.0044}$ | $0.7449_{\pm 0.0044}$ | $0.6592_{\pm 0.0051}$ | $0.3636_{\pm 0.0343}$ | $0.6707_{\pm 0.0368}$ | $0.6402_{\pm 0.0376}$ | $0.8173_{\pm 0.0106}$ |
| AuxLossFree+z | Ling-mini-2.0 | $0.6878_{\pm 0.0036}$ | $0.4480_{\pm 0.0044}$ | $0.7396_{\pm 0.0044}$ | $0.6520_{\pm 0.0054}$ | $0.3535_{\pm 0.0341}$ | $0.6500_{\pm 0.0214}$ | $0.6220_{\pm 0.0380}$ | $0.7885_{\pm 0.0112}$ |
| AuxLossFree+o+v | | $0.7119_{\pm 0.0036}$ | $0.4892_{\pm 0.0044}$ | $0.7596_{\pm 0.0044}$ | $0.7016_{\pm 0.0048}$ | $0.3838_{\pm 0.0346}$ | $0.6620_{\pm 0.0212}$ | $0.6524_{\pm 0.0373}$ | $0.8309_{\pm 0.0103}$ |
| **AuxLossFree+cp+sp** | | $\mathbf{0.7667}_{\pm 0.0036}$ | $\mathbf{0.5002}_{\pm 0.0044}$ | $\mathbf{0.7627}_{\pm 0.0042}$ | $\mathbf{0.7269}_{\pm 0.0048}$ | $\mathbf{0.3889}_{\pm 0.0347}$ | $\mathbf{0.6820}_{\pm 0.0208}$ | $\mathbf{0.6890}_{\pm 0.0363}$ | $\mathbf{0.8559}_{\pm 0.0101}$ |

*Table 3.* Evaluation score on Qwen3-30B-A3B-Instruct-2507 fine-tuning tasks. The last four rows stands for the performance for mmlu dataset with different domains.

| Dataset | Metric | $\mathcal{L}_{lb}$ | $\mathcal{L}_{lb,sp,cp}$ |
|---|---|---|---|
| openai_humaneval | humaneval_pass@1 | 92.07 | **95.73** |
| gsm8k | accuracy | 93.33 | **94.16** |
| math_prm800k_500 | accuracy | 94.00 | **94.20** |
| mmlu | naive_average | 78.97 | **79.86** |
| mmlu-weighted | weighted_average | 76.35 | **77.10** |

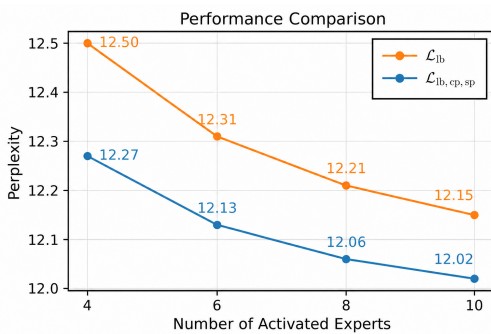

*Figure 5.* Scalability performance with varying number of activated experts ($N$) on medium-sized models.

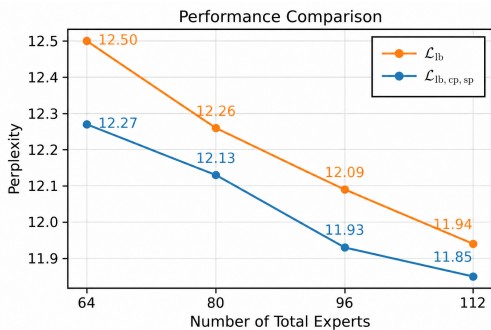

*Figure 6.* Scalability performance with varying total number of experts ($E$) on medium-sized models.

cp + sp provides the most reliable and across-the-board improvements and remains stronger than the variance-based auxiliary, indicating that activation/path-based regularization transfers more robustly in SFT than directly maximizing routing-logit variance.

**Full-Parameter Fine-Tuning Evaluation (Qwen3-30B-A3B).** We consider the fine-tuning tasks under `Qwen3-30B-A3B-Instruct-2507` model on an internal corpus of 38B tokens (see details in Appendix E). We evaluate on a broad suite of reasoning and knowledge-intensive benchmarks, including HumanEval (Chen et al., 2021), GSM8K (Cobbe et al., 2021), math500_PRM800K_dataset (Lightman et al., 2023), and MMLU (Hendrycks et al., 2020). Across nearly all settings, incorporating $\mathcal{R}_{cp}$ and $\mathcal{R}_{sp}$ outperforms the baseline, yielding consistent gains on reasoning-oriented tasks as well as aggregate knowledge measures as Table 3. While a minor fluctuation is observed on the humanities subset of MMLU, the overall trend remains positive, confirming that our objectives not only sharpen specialization in pre-training but also transfer effectively to finetuning adaptation.

**Scalability of the auxiliary loss.** We evaluate scalability on *medium-sized* MoE models by varying (i) the number of activated experts ($N$) and (ii) the total number of experts ($E$). As shown in Figures 5 and 6, our auxiliary objectives consistently achieve lower perplexity than the load-balance-only baseline across both axes. Notably, with our loss, activating

only $N$=6 experts already outperforms the baseline even with $N$=10 (12.13 vs. 12.15), and using a smaller expert pool $E$=96 surpasses the baseline with $E$=112 (11.93 vs. 11.94), indicating improved scaling efficiency with fewer active/total experts.

## 8. Conclusion

We presented two plug-and-play losses that directly optimize expert specialization in MoE models. The intra-layer specialization loss ($\mathcal{R}_{sp}$) penalizes activation similarity between experts processing identical tokens, while the cross-

layer coupling loss ($\mathcal{R}_{cp}$) maximizes joint routing probabilities across adjacent layers to establish coherent expert pathways. These losses require no architectural modifications, integrate seamlessly with existing objectives, and are theoretically grounded. Empirically, our approach improves performance across all tested scales and MoE variants while increasing inference throughput through stable expert paths.

## Acknowledgments

This work is funded by the National Natural Science Foundation of China (No. W2441021, 12301392, 92370121, 12288101), and the National Key Research and Development Program of China (No. 2024YFA1012902). This research is also supported by Zhejiang Lab and the AI for Science Institute, Beijing, China.

## Impact Statement

This work focuses on improving our basic understanding of machine learning methods. Although the ideas presented here could have broader practical effects, we are not aware of any direct or significant societal impacts that require specific mention at this time.

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

# Appendix

## A. Additional related works

Here we present some additional related works that have not been discussed in Section 2.

**Cross-layer signals and information.** Recent work observes that routing decisions and activations exhibit strong cross-layer correlations. Read-ME precomputes routing decisions across layers and leverages inter-layer expert affinity to optimize scheduling and caching (Cai et al., 2024; Yao et al., 2024). Meanwhile, other studies demonstrate that inter-layer residuals—especially when exhibiting low-rank or redundant structures—can be harnessed to improve efficiency (Liu et al., 2024b; Pagliardini et al., 2024; Kong et al., 2025; Zhang et al., 2025). These methods demonstrate that cross-layer structure is a robust empirical phenomenon, but they mainly use it for system or training efficiency rather than as an explicit learning signal. In contrast, our cross-layer coupling loss turns inter-layer routing affinity into a supervised objective: it encourages tokens to follow coherent expert paths through depth, thereby reinforcing intra-layer specialization while remaining compatible with standard load-balancing objectives.

## B. Proofs for Gradient Decorrelation and Cross-Layer Specialization Propagation.

In this section, we present the proofs for the proposed propositions in Section 4 and 5.

### B.1. Proof of Proposition 4.1

**Proposition B.1** (Proposition 4.1). *For any two activated experts $e, \nu \in \mathbb{A}_i^{(\ell)}$, the cosine similarity between the gradients of the total loss $\mathcal{L}$ with respect to their down-projection matrices satisfies*

$$\cos\left( \frac{\partial \mathcal{L}}{\partial W_{\text{down}}^{(\ell,e)}}, \frac{\partial \mathcal{L}}{\partial W_{\text{down}}^{(\ell,\nu)}} \right) = \cos\left( z_i^{(\ell,e)}, z_i^{(\ell,\nu)} \right), \tag{9}$$

*where $z_i^{(\ell,e)}$ and $z_i^{(\ell,\nu)}$ are the corresponding intermediate activations.*

*Proof.* As the routing weights do not affect the cosine, without loss of generality we assume that the activated experts contribute with equal weights. Then the output of MoE blocks for layer $\ell$ can be written as

$$E(x_i^{(\ell)}) := \sum_{e \in \mathbb{A}_i^{(\ell)}} y_i^{(\ell,e)}. \tag{10}$$

Then for any $e \in \mathbb{A}_i^{(\ell)}$ it holds that

$$\frac{\partial \mathcal{L}}{\partial y^{(\ell,e)}} = \frac{\partial \mathcal{L}}{\partial E(x_i^{(\ell)})}. \tag{11}$$

As $y_i^{(\ell,e)} = z_i^{(\ell,e)} W_{\text{down}}^{(\ell)}$, thus from (11) it comes for any $e \in \mathbb{A}_i^{(\ell)}$ that:

$$\frac{\partial \mathcal{L}}{\partial W_{\text{down}}^{(\ell,e)}} = \frac{\partial \mathcal{L}}{\partial y_i^{(\ell,e)}} \frac{\partial y_i^{(\ell,e)}}{\partial W_{\text{down}}^{(\ell,e)}} = \frac{\partial \mathcal{L}}{\partial E(x_i^{(\ell)})} z_i^{(\ell,e)}. \tag{12}$$

Using the Frobenius inner-product identity $\langle ab^\top, \ cd^\top \rangle_F = (a^\top c)(b^\top d)$ and $\|ab^\top\|_F = \|a\|_2 \|b\|_2$, we obtain that for

$e_1, e_2 \in \mathbb{A}_i^{(\ell)}$ it holds that

$$
\cos\left(\frac{\partial \mathcal{L}}{\partial W_{\text{down}}^{(\ell,e_1)}}, \frac{\partial \mathcal{L}}{\partial W_{\text{down}}^{(\ell,e_2)}}\right) = \frac{\left[\left(z_i^{(\ell,e_1)}\right)^\top z_i^{(\ell,e_2)}\right] \cdot \left[\frac{\partial \mathcal{L}}{\partial \left(y_i^{(\ell,e)}\right)^\top} \frac{\partial \mathcal{L}}{\partial y_i^{(\ell,e)}}\right]}{\left\|z_i^{(\ell,e_1)}\right\|_2 \cdot \left\|\frac{\partial \mathcal{L}}{\partial \left(y_i^{(\ell,e)}\right)}\right\|_2 \cdot \left\|z_i^{(\ell,e_2)}\right\|_2 \cdot \left\|\frac{\partial \mathcal{L}}{\partial \left(y_i^{(\ell,e)}\right)}\right\|_2}
$$

$$
= \frac{z_i^{(\ell,e_1)}\left(z_i^{(\ell,e_2)}\right)^\top}{\left\|z_i^{(\ell,e_1)}\right\|_2 \left\|z_i^{(\ell,e_2)}\right\|_2} = \cos\left(z_i^{(\ell,e_1)}, z_i^{(\ell,e_2)}\right). \tag{13}
$$

When condisering the case that each expert output is scaled by a positive routing weight, i.e., $\widetilde{y}_i^{(\ell,e)} = \alpha_i^{(\ell,e)} \cdot z_i^{(\ell,e)} W_{\text{down}}^{(\ell,e)}$, where $\alpha_i^{(\ell,e)} \in (0,1]$ is the routing weight. Similar to (12), we can obtain that

$$
\frac{\partial \mathcal{L}}{\partial W_{\text{down}}^{(\ell,e)}} = \alpha_i^{(\ell,e)} \cdot \frac{\partial \mathcal{L}}{\partial E(x_i^{(\ell)})} z_i^{(\ell,e)}.
$$

Thus the common positive factor cancels in the cosine similarity, leaving the result unchanged. $\square$

## B.2. Proof of Proposition 5.1

**Proposition B.2** (Proposition 5.1). *Let $\mathbb{A}_i^{(\ell)}$ denote the set of activated experts for token $x_i$ at layer $\ell$. Consider adjacent layers $\ell$ and $\ell + 1$ satisfying:*

*1. **Representation continuity.** For a token $x_i$, its representations evolve smoothly across layers: $\cos(x_i^{(\ell)}, x_i^{(\ell+1)}) \geq 1 - \delta^2$ for small $\delta \in (0,1)$.*

*2. **Source-layer specialization.** Layer $\ell$ exhibits expert specialization with nearly orthogonal router weights: for experts $e_1 \in \mathbb{A}_i^{(\ell)}$ and $e_2 \in \mathbb{A}_j^{(\ell)}$ processing different tokens $x_i \neq x_j$, we have $|\cos(r^{(\ell,e_1)}, r^{(\ell,e_2)})| \leq \varepsilon$ for a small $\varepsilon \in (0,1)$.*

*3. **Strong cross-layer coupling.** Adjacent layers exhibit stable expert pathways with high routing correlation. For any expert $e \in \mathbb{A}_i^{(\ell)}$ activated by token $x_i$, there exists a corresponding expert $\nu \in \mathbb{A}_i^{(\ell+1)}$ such that both routing decisions are confident: $\cos(x_i^{(\ell)}, r^{(\ell,e)}) \geq 1 - \iota^2$ and $\cos(x_i^{(\ell+1)}, r^{(\ell+1,\nu)}) \geq 1 - \iota^2$ for small $\iota \in (0,1)$.*

*Under these conditions, layer $\ell + 1$ inherits the specialization structure from layer $\ell$:*

$$
\left|\cos\left(r^{(\ell+1,\nu_1)}, r^{(\ell+1,\nu_2)}\right)\right| \leq \varepsilon + O(\delta, \iota) \tag{14}
$$

*for experts $\nu_1 \in \mathbb{A}_i^{(\ell+1)}$ and $\nu_2 \in \mathbb{A}_j^{(\ell+1)}$ processing different tokens, where the error term $O(\delta, \iota)$ vanishes as $\delta$ and $\iota$ decrease to $0$.*

*Proof.* From the first Assumption, It can be obtained that:

$$
\left\|\frac{x_i^{(\ell,e)}}{\left\|x_i^{(\ell,e)}\right\|} - \frac{x_i^{(\ell+1,e)}}{\left\|x_i^{(\ell+1,e)}\right\|}\right\|^2 = 2 - 2\cos\left(x_i^{(\ell,e)}, x_i^{(\ell+1,e)}\right) \leq 2\delta^2. \tag{15}
$$

Similarly, it holds that:

$$
\left\|\frac{x_i^{(\ell,e)}}{\left\|x_i^{(\ell,e)}\right\|} - \frac{r^{(\ell,e)}}{\left\|r^{(\ell,e)}\right\|}\right\|^2 \leq 2\iota^2, \quad \left\|\frac{x_i^{(\ell+1,\nu)}}{\left\|x_i^{(\ell+1,\nu)}\right\|} - \frac{r^{(\ell+1,\nu)}}{\left\|r^{(\ell+1,\nu)}\right\|}\right\|^2 \leq 2\iota^2. \tag{16}
$$

Then from Eq. (15) and Eq. (16), it holds that

$$
\begin{aligned}
&\left\| \frac{r^{(\ell,e)}}{\left\|r^{(\ell,e)}\right\|} - \frac{r^{(\ell+1,\nu)}}{\left\|r^{(l+1,\nu)}\right\|} \right\| \\
&\leq \left\| \frac{x_i^{(\ell,e)}}{\left\|x_i^{(\ell,e)}\right\|} - \frac{x_i^{(\ell+1,e)}}{\left\|x_i^{(\ell+1,e)}\right\|} \right\| + \left\| \frac{x_i^{(\ell,e)}}{\left\|x_i^{(\ell,e)}\right\|} - \frac{r^{(\ell,e)}}{\left\|r^{(\ell,e)}\right\|} \right\| + \left\| \frac{x_i^{(\ell+1,\nu)}}{\left\|x_i^{(\ell+1,\nu)}\right\|} - \frac{r^{(\ell+1,\nu)}}{\left\|r^{(\ell+1,\nu)}\right\|} \right\| \\
&\leq \sqrt{2}\left(\delta + 2\iota\right).
\end{aligned}
\tag{17}
$$

Then it holds that

$$
\cos\left(r^{(\ell,e)}, r^{(\ell+1,\nu)}\right) = 1 - \frac{1}{2}\left\| \frac{r^{(\ell,e)}}{\left\|r^{(\ell,e)}\right\|} - \frac{r^{(\ell+1,\nu)}}{\left\|r^{(\ell+1,\nu)}\right\|} \right\|^2 \geq 1 - (\delta + 2\iota)^2.
\tag{18}
$$

Then we prove (6). Let

$$
\begin{aligned}
\tilde{r}^{(\ell,e_1)} &:= \frac{r^{(\ell,e_1)}}{\left\|r^{(\ell,e_1)}\right\|}, \quad \tilde{r}^{(\ell+1,\nu_1)} := \frac{r^{(\ell+1,\nu_1)}}{\left\|r^{(\ell+1,\nu_1)}\right\|}, \\
\tilde{r}^{(\ell,e_2)} &:= \frac{r^{(\ell,e_2)}}{\left\|r^{(\ell,e_2)}\right\|}, \quad \tilde{r}^{(\ell+1,\nu_2)} := \frac{r^{(\ell+1,\nu_2)}}{\left\|r^{(\ell+1,\nu_2)}\right\|}.
\end{aligned}
$$

Then it comes that:

$$
\begin{aligned}
&\left| \left\langle \tilde{r}^{(\ell+1,\nu_1)}, \tilde{r}^{(\ell+1,\nu_2)} \right\rangle \right| \\
&= \left| \left\langle \tilde{r}^{(\ell,e_1)}, \tilde{r}^{(\ell,e_2)} \right\rangle \right| + \left| \left\langle \tilde{r}^{(\ell,e_1)} - \tilde{r}^{(\ell+1,\nu_1)}, \tilde{r}^{(\ell,e_2)} \right\rangle \right| + \left| \left\langle \tilde{r}^{(\ell,e_1)}, \tilde{r}^{(\ell,e_2)} - \tilde{r}^{(\ell+1,\nu_2)} \right\rangle \right| \\
&\quad + \left| \left\langle \tilde{r}^{(\ell,e_1)} - \tilde{r}^{(\ell+1,\nu_1)}, \tilde{r}^{(\ell,e_2)} - \tilde{r}^{(\ell+1,\nu_2)} \right\rangle \right| \\
&\leq \varepsilon + 2\sqrt{2}\left(\delta + 2\iota\right) + 2\left(\delta + 2\iota\right)^2,
\end{aligned}
\tag{19}
$$

where the last inequality is from (17). Then we finish the proof of this lemma. $\qquad\square$

## C. Theory Details for Specialization, Routing Sharpness, and Cross-Layer Coupling

In this section, we aim to establish a comprehensive theoretical framework for the proposed specialization and coupling loss.

**Notations.** Throughout this appendix we consider a single MoE layer with $E \geq 2$ experts and **top-1 routing** ($k = 1$) at inference. Let $(t, y) \sim D$ be the data distribution. Each expert $e$ incurs per-token loss

$$
\ell_e(t) := \ell\big(f_e(t), y\big),
$$

and the router produces logits $z(t) = (z_1(t), \ldots, z_E(t))$ with softmax probabilities

$$
g_e(t) = \frac{\exp(z_e(t))}{\sum_{j=1}^{E} \exp(z_j(t))}, \qquad e \in \{1, \ldots, E\}.
$$

The standard soft-routing objective is the mixture loss

$$
L(t) := \sum_{e=1}^{E} g_e(t)\, \ell_e(t).
$$

When the best expert is unique, we write $e^\star(t) := \arg\min_e \ell_e(t)$ and define the (token-wise) margin

$$
\Delta(t) := \min_{e \neq e^\star(t)} \big(\ell_e(t) - \ell_{e^\star(t)}(t)\big) \in [0, \infty).
$$

**Section Roadmap.** Section 6 in the main text presents a lightweight theoretical picture of how $\mathcal{R}_{\mathrm{sp}}$ and $\mathcal{R}_{\mathrm{cp}}$ interact with routing and load balancing. This section provides the formal counterparts: additional definitions, auxiliary quantities, and proofs that support the claims summarized in Section 6. Concretely:

- Appendix C.1 formalizes the specialization–routing mutual reinforcement mechanism (weak loss advantage ⇒ decisive routing, and decisive routing ⇒ specialization amplification), and derives an explicit entropy corollary.

- Appendix C.2 models cross-layer coupling as a learning signal, defines a permutation-invariant coupling coefficient, and proves a backward-transfer guarantee under strong coupling.

- Appendix C.3 provides sufficient conditions and constructions establishing compatibility with standard load balancing.

### C.1. Specialization–Routing Mutual Reinforcement

C.1.1. WEAK SPECIALIZATION ⇒ DECISIVE ROUTING

**Theorem C.1** (Weak specialization implies high-probability decisive routing). *We assume that*

*(A1) (**Local uniqueness**) For D-almost every token $t$, the best expert $e^\star(t)$ is unique.*

*(A2) (**Weak specialization: high-probability margin**) There exist $\gamma_0 > 0$ and $\varepsilon_0 \in [0, 1)$ such that*

$$\mathbb{P}_{(t,y)\sim D}\big[\Delta(t) \geq \gamma_0\big] \ \geq \ 1 - \varepsilon_0. \tag{20}$$

*Consider a router update in which $\{\ell_e(t)\}_{e=1}^E$ are treated as locally constant w.r.t. the router logits $z(t)$. Then:*

*1. For any token $t$ with $\Delta(t) > 0$ and $g_{e^\star(t)}(t) < 1$,*

$$\frac{\partial L(t)}{\partial z_{e^\star(t)}(t)} \ < \ 0. \tag{21}$$

   *Consequently, a gradient-descent update increases $z_{e^\star(t)}(t)$.*

*2. For any $\delta \in (0, 1)$,*

$$\mathbb{P}_{(t,y)\sim D}\big[g_{e^\star(t)}(t) \geq 1 - \delta\big] \ \geq \ 1 - \varepsilon_0 - \frac{\mathbb{E}_{(t,y)\sim D}\big[L(t) - \ell_{e^\star(t)}(t)\big]}{\gamma_0\,\delta}. \tag{22}$$

*Proof.* Fix a token $t$ and abbreviate $\ell_e = \ell_e(t)$, $g_e = g_e(t)$, $z_e = z_e(t)$, and $L = L(t)$. For the softmax, $\frac{\partial g_j}{\partial z_e} = g_j(\mathbf{1}\{j = e\} - g_e)$, hence

$$\frac{\partial L}{\partial z_e} = \sum_{j=1}^E \ell_j \frac{\partial g_j}{\partial z_e} = \sum_{j=1}^E \ell_j g_j(\mathbf{1}\{j = e\} - g_e) = \ell_e g_e - g_e \sum_{j=1}^E g_j \ell_j = g_e(\ell_e - L). \tag{23}$$

Let $e^\star = e^\star(t)$. If $\Delta(t) > 0$ and $g_{e^\star} < 1$, then there exists at least one $e \neq e^\star$ with $g_e > 0$ and $\ell_e > \ell_{e^\star}$. As $L = \sum_j g_j \ell_j$ is a convex combination with positive mass on a value strictly larger than $\ell_{e^\star}$, we have $L > \ell_{e^\star}$. Therefore $\frac{\partial L}{\partial z_{e^\star}} = g_{e^\star}(\ell_{e^\star} - L) < 0$, proving (21).

For the high-probability sharpness bound, on any token $t$ where $e^\star(t)$ is unique and $\Delta(t) > 0$,

$$L(t) - \ell_{e^\star(t)}(t) = \sum_{e \neq e^\star(t)} g_e(t)\big(\ell_e(t) - \ell_{e^\star(t)}(t)\big) \geq \sum_{e \neq e^\star(t)} g_e(t)\,\Delta(t) = \Delta(t)\big(1 - g_{e^\star(t)}(t)\big), \tag{24}$$

which implies

$$1 - g_{e^\star(t)}(t) \ \leq \ \frac{L(t) - \ell_{e^\star(t)}(t)}{\Delta(t)}. \tag{25}$$

Let $G := \{\Delta(t) \geq \gamma_0\}$. By (20), $\mathbb{P}(G^c) \leq \varepsilon_0$, where $G^c$ denotes the complement of event. Then for any $\delta \in (0,1)$,

$$\mathbb{P}\big(g_{e^\star(t)}(t) < 1 - \delta\big) = \mathbb{P}\big(1 - g_{e^\star(t)}(t) > \delta\big) \leq \mathbb{P}(G^c) + \mathbb{P}\big(1 - g_{e^\star(t)}(t) > \delta, \, G\big), \tag{26}$$

For the event $G$, inequality (25) yields $1 - g_{e^\star(t)}(t) \leq \dfrac{L(t) - \ell_{e^\star(t)}(t)}{\gamma_0}$, then we can obtain

$$\mathbb{P}\big(1 - g_{e^\star(t)}(t) > \delta, \, G\big) \leq \mathbb{P}\left(\frac{L(t) - \ell_{e^\star(t)}(t)}{\gamma_0} > \delta\right) = \mathbb{P}\big(L(t) - \ell_{e^\star(t)}(t) > \gamma_0\delta\big). \tag{27}$$

Applying Markov's inequality to the nonnegative random variable $L(t) - \ell_{e^\star(t)}(t)$ gives

$$\mathbb{P}\big(L(t) - \ell_{e^\star(t)}(t) > \gamma_0\delta\big) \leq \frac{\mathbb{E}\big[L(t) - \ell_{e^\star(t)}(t)\big]}{\gamma_0\delta}. \tag{28}$$

Combining the last three displays and using $\mathbb{P}(G^c) \leq \varepsilon_0$ yields

$$\mathbb{P}\big(g_{e^\star(t)}(t) \geq 1 - \delta\big) \geq 1 - \varepsilon_0 - \frac{\mathbb{E}\big[L(t) - \ell_{e^\star(t)}(t)\big]}{\gamma_0\delta},$$

which proves (22). $\qquad\square$

Theorem C.1 formalizes a simple but important point: the standard MoE mixture objective already prefers *decisive* routing. As soon as one expert attains even a weak per-token loss advantage, the router gradient increases that expert's logit, so probability mass moves toward the best expert. Moreover, if the mixture loss stays close to the best-expert loss on average (a small oracle gap), then the router must place nearly all mass on the best expert for most tokens, yielding low-entropy routing. This is exactly the direction in which $\mathcal{R}_{\mathrm{sp}}$ helps: by discouraging expert overlap, it makes such loss advantages more persistent and easier to amplify.

*Remark* C.2 (Why is the oracle gap typically small during training?). Let $G(t) := L(t) - \ell_{e^\star(t)}(t) \geq 0$ denote the *oracle gap* between the router's soft mixture loss $L(t)$ and the best-expert loss $\ell_{e^\star(t)}(t)$ at token $t$. In practice, $G(t)$ tends to shrink during training for two coupled reasons. First, router updates increase the logit (and hence probability) of the current best expert, which decreases the mixture loss $L(t)$. Second, expert updates reduce $\ell_e(t)$ on the tokens they repeatedly receive, which makes best-expert advantages more pronounced over time. Once experts become even mildly differentiated, this interaction naturally drives routing to become more decisive, pushing $L(t)$ closer to the oracle loss.

We next convert the high-probability sharpness event in Theorem C.1 into an explicit upper bound on router entropy, which serves as a convenient routing-clarity diagnostic.

**Corollary C.3** (Entropy bound as a consequence of decisive routing). *Let $H(g(t)) := -\sum_{e=1}^{E} g_e(t) \log g_e(t)$. For any $\delta \in (0, 1/2]$, on the event $\{g_{e^\star(t)}(t) \geq 1 - \delta\}$ we have*

$$H(g(t)) \leq h(\delta) + \delta \log(E - 1), \qquad h(\delta) := -\delta \log \delta - (1 - \delta) \log(1 - \delta). \tag{29}$$

*Consequently, with the same lower bound as in* (22),

$$\mathbb{P}_{(t,y)\sim D}[H(g(t)) \leq h(\delta) + \delta \log(E - 1)] \geq 1 - \varepsilon_0 - \frac{\mathbb{E}_{(t,y)\sim D}\big[L(t) - \ell_{e^\star(t)}(t)\big]}{\gamma_0\,\delta}. \tag{30}$$

*Proof.* Fix a token $t$ and let $p := g_{e^\star(t)}(t)$. Consider all probability vectors $g \in \Delta^{E-1}$ with $g_{e^\star} = p$. Among these, the Shannon entropy $H(g)$ is maximized when the remaining mass $1 - p$ is spread uniformly over the other $E - 1$ coordinates, i.e., $g_e = \dfrac{1-p}{E-1}$ for all $e \neq e^\star$ (this follows from the strict concavity of $x \mapsto -x \log x$ and Jensen's inequality). Therefore,

$$H(g(t)) \leq -p \log p - \sum_{e \neq e^\star} \frac{1-p}{E-1} \log \frac{1-p}{E-1} = -p \log p - (1-p) \log \frac{1-p}{E-1}. \tag{31}$$

Rewriting gives

$$H(g(t)) \leq \underbrace{\left( -p \log p - (1-p)\log(1-p) \right)}_{= h(1-p)} + (1-p)\log(E-1) = h(1-p) + (1-p)\log(E-1). \tag{32}$$

As $p \geq 1 - \delta$, we have $1 - p \leq \delta \leq 1/2$, and since $h(\cdot)$ is nondecreasing on $[0, 1/2]$, it follows that $h(1-p) \leq h(\delta)$ and $(1-p)\log(E-1) \leq \delta \log(E-1)$. This proves (29). The probability statement follows by observing that the event $\{g_{e^\star(t)}(t) \geq 1 - \delta\}$ implies the event $\{H(g(t)) \leq h(\delta) + \delta \log(E-1)\}$ and applying (22). $\qquad\square$

Corollary C.3 converts the sharpness event $g_{e^\star}(t) \geq 1 - \delta$ into an explicit upper bound on the router entropy. Combined with Theorem C.1, it provides a clean "loss margin $\Rightarrow$ low entropy" link without introducing any extra entropy regularizer.

### C.1.2. DECISIVE ROUTING $\Rightarrow$ SPECIALIZATION AMPLIFICATION

We now formalize the reverse direction: *if routing is sufficiently clear and aligned, each expert trains on a cleaner distribution concentrated on tokens where it already has advantage, which forces its performance on that region to improve.* This result is stated in a deliberately weak (but assumption-light) form that captures the positive feedback mechanism.

Let $\{S_e^\star\}_{e=1}^E$ be measurable subsets of $\mathcal{T}$ that partition the token space up to $D$-null sets (i.e., $D(\cup_e S_e^\star) = 1$ and $D(S_e^\star \cap S_{e'}^\star) = 0$ for $e \neq e'$), interpreted as the current "advantage regions". Given router weights $g(e \mid t)$ and data distribution $D$, define the *effective training distribution* seen by expert $e$:

$$D_e(A) := \frac{\mathbb{E}_{(t,y)\sim D}\left[g(e \mid t)\mathbf{1}\{(t,y) \in A\}\right]}{\mathbb{E}_{(t,y)\sim D}[g(e \mid t)]}, \qquad A \subseteq \mathcal{T} \times \mathcal{Y}. \tag{33}$$

Define the *purity* of expert $e$ as

$$\alpha_e := \mathbb{P}_{(t,y)\sim D_e}\left[t \in S_e^\star\right] = D_e\left(S_e^\star \times \mathcal{Y}\right). \tag{34}$$

Finally define the effective risk and the region-conditional risk:

$$R_e^{\text{eff}}(f_e) := \mathbb{E}_{(t,y)\sim D_e}\left[\ell(f_e(t), y)\right], \qquad R_e^S(f_e) := \mathbb{E}_{(t,y)\sim D_e}\left[\ell(f_e(t), y) \mid t \in S_e^\star\right].$$

**Theorem C.4** (Clear routing improves region-conditional risk (a weak specialization amplifier))**.** *Fix an expert $e$ and assume:*

*(B1)* *(**High purity**) $\alpha_e \geq 1 - \eta$ for some $\eta \in (0, 1)$.*

*(B2)* *(**Effective optimization**) After a training phase, expert $e$ achieves*

$$R_e^{\text{eff}}(f_e^{\text{new}}) \leq R_e^{\text{eff}}(f_e^{\text{old}}) - \Delta_e \qquad \text{for some } \Delta_e > 0. \tag{35}$$

*(B3)* *(**Bounded loss**) $0 \leq \ell(\hat{y}, y) \leq B$ almost surely for some $B > 0$.*

*Then the region-conditional risk on the advantage region satisfies*

$$R_e^S(f_e^{\text{new}}) \leq R_e^S(f_e^{\text{old}}) - (\Delta_e - \eta B). \tag{36}$$

*In particular, if $\Delta_e > \eta B$ and $\eta$ is small (clear, well-aligned routing), then expert $e$ must strictly improve on its advantage region $S_e^\star$ under $D_e(\cdot \mid t \in S_e^\star)$.*

*Proof.* Under $D_e$, decompose the effective risk by conditioning on whether $t \in S_e^\star$:

$$R_e^{\text{eff}}(f_e) = \mathbb{E}_{D_e}\left[\ell(f_e(t), y)\right] = \alpha_e \mathbb{E}_{D_e}\left[\ell(f_e(t), y) \mid t \in S_e^\star\right] + (1 - \alpha_e)\mathbb{E}_{D_e}\left[\ell(f_e(t), y) \mid t \notin S_e^\star\right]. \tag{37}$$

Let $R_e^{\bar{S}}(f_e) := \mathbb{E}_{D_e}[\ell(f_e(t), y) \mid t \notin S_e^\star]$. Define the changes

$$\Delta R^{\text{eff}} := R_e^{\text{eff}}(f_e^{\text{new}}) - R_e^{\text{eff}}(f_e^{\text{old}}), \qquad \Delta R^S := R_e^S(f_e^{\text{new}}) - R_e^S(f_e^{\text{old}}), \qquad \Delta R^{\bar{S}} := R_e^{\bar{S}}(f_e^{\text{new}}) - R_e^{\bar{S}}(f_e^{\text{old}}). \tag{38}$$

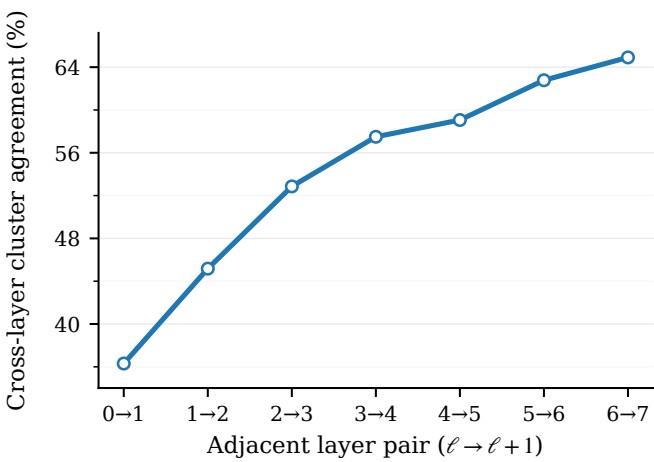

*Figure 7.* Cross-layer cluster agreement (%) for adjacent layer pairs ($\ell \rightarrow \ell+1$) under top-1 routing.

By the above decomposition applied to $f_e^{\text{new}}$ and $f_e^{\text{old}}$ and subtracting, we get

$$\Delta R^{\text{eff}} = \alpha_e \, \Delta R^S + (1 - \alpha_e) \, \Delta R^{\bar{S}}. \tag{39}$$

Assumption (35) gives $\Delta R^{\text{eff}} \leq -\Delta_e$. By boundedness (B3), both $R_e^{\bar{S}}(f_e^{\text{new}})$ and $R_e^{\bar{S}}(f_e^{\text{old}})$ lie in $[0, B]$, hence $\Delta R^{\bar{S}} \leq B$. Therefore,

$$\alpha_e \, \Delta R^S = \Delta R^{\text{eff}} - (1 - \alpha_e)\Delta R^{\bar{S}} \leq -\Delta_e + (1 - \alpha_e)B. \tag{40}$$

Using (B1), $1 - \alpha_e \leq \eta$ and $\alpha_e \leq 1$, yielding

$$\Delta R^S \leq -\Delta_e + \eta B. \tag{41}$$

Rearranging gives (36). □

Theorem C.4 captures the reverse direction of the loop. Once routing is sharp and aligned, each expert's effective training distribution becomes *high-purity*: it is dominated by the subset of tokens where that expert already has comparative advantage. Under this higher-purity data distribution, progress on the effective risk must translate into progress on the expert's preferred region, so the expert's regional advantage widens over time. In short, decisive routing is not only about low entropy—it is a data-selection mechanism that reduces gradient interference and *amplifies* functional specialization.

### C.2. Cross-layer coupling as a learning signal

So far we focused on the within-layer feedback mechanism; we next show how cross-layer coupling provides an additional learning signal that stabilizes and propagates these effects across depth.

**Observation: cross-layer coupling.** Recent work has identified an emergent property in MoE training known as *cross-layer coupling* (Cai et al., 2024; Yao et al., 2024): the expert activated at layer $\ell$ strongly predicts the expert activated at layer $\ell+1$. Empirically, this phenomenon is visible early in training and becomes increasingly pronounced as optimization proceeds . These observations suggest that coupling is not merely an artifact of optimization, but reflects a meaningful structural regularity in how tokens are routed through depth.

**A token-distribution prior view.** We consider the **top-1 routing setting**, i.e., each token activates exactly one expert per layer. We provide an intuitive explanation for why such coupling naturally arises. Tokens are not uniformly distributed in semantic space; rather, they exhibit latent structure . For example, some tokens predominantly encode mathematical reasoning patterns, others linguistic syntax, and others domain-specific factual knowledge. This induces a *latent, relatively good token-to-expert allocation*: a partition of the token distribution into subsets for which different experts are more suitable .

Importantly, we do *not* assume this allocation is globally optimal or explicitly known. Instead, it should be viewed as a soft prior induced by the data distribution itself. During training, routers at different layers implicitly attempt to approximate this same underlying allocation. Cross-layer coupling emerges when multiple layers align to this shared token-distribution prior, up to permutations of expert identities .

**Empirical justification of the prior.** To justify the token-distribution prior, we approximate it by a natural partition of the pre-router token distribution at each layer and evaluate how consistent these partitions are across depth. Concretely, we use a 0.4B MoE with total experts $E=8$ and activated experts $k=1$ per layer. At every layer $\ell$, we run $k$-means to cluster tokens into $E$ groups and apply a distance-greedy balancing step to equalize cluster sizes. Because cluster indices are arbitrary per layer, we align clusters between layers $\ell$ and $\ell+1$ using the Hungarian algorithm (maximum bipartite matching over the inter-layer confusion matrix). We report the percentage of tokens whose cluster at layer $\ell$ matches the aligned cluster at layer $\ell+1$; the partition difference is 1 minus this agreement (Fig. 7).

**Why coupling strengthens with depth.** A key empirical observation is that cross-layer coupling becomes stronger in deeper layers . (See Fig. 8, where deeper-layer pairs exhibit sharper coupling patterns .) Our modeling provides a natural explanation for this phenomenon.

Let $h_\ell(t) \in \mathbb{R}^{d_\ell}$ denote the representation of token $t$ at layer $\ell$. For a given latent token-to-expert assignment $A(t)$ induced by the data distribution, define the best achievable linear routing error at layer $\ell$ as

$$\alpha_\ell := \inf_{W_\ell \in \mathbb{R}^{E \times d_\ell}} \mathbb{P}_{t \sim D}\left[\arg\max_e \langle W_{\ell,e}, h_\ell(t)\rangle \neq A(t)\right].$$

In deep networks, representations typically become more linearly separable with depth (Zhang et al., 2023), suggesting $\alpha_{\ell+1} \leq \alpha_\ell$ in many regimes.

Since standard MoE routers are approximately linear classifiers on $h_\ell(t)$, their routing error can be decomposed into: (i) a *representation-induced approximation error* (captured by $\alpha_\ell$), and (ii) an *optimization error* due to imperfect training. As depth increases, the approximation error decreases, allowing later-layer routers to more faithfully align with the latent token allocation. As a result, adjacent layers increasingly agree on routing decisions, leading to stronger cross-layer coupling in deeper parts of the network.

**Using later-layer routing as a learning signal.** The discussion above suggests that, on average, routing decisions at layer $\ell + 1$ are a more accurate proxy of the latent token allocation than those at layer $\ell$. This motivates a simple but powerful idea: *use routing decisions from deeper layers as a self-supervised target to guide earlier layers*. By encouraging adjacent layers to agree on expert identities, we can propagate improved routing structure backward through depth, reducing routing ambiguity and accelerating specialization.

**Coupling coefficient and backward transfer guarantee.** For top-1 routing, let

$$C_\ell(t) \in \{1, \ldots, E\}, \qquad C_{\ell+1}(t) \in \{1, \ldots, E\}$$

denote the selected experts at layers $\ell$ and $\ell + 1$ for token $t$. We quantify cross-layer pathway consistency using the permutation-invariant coupling coefficient

$$\kappa_{\ell \to \ell+1} := \max_{\pi \in S_E} \mathbb{P}_{t \sim D}\left[\pi\big(C_\ell(t)\big) = C_{\ell+1}(t)\right].$$

A value of $\kappa_{\ell \to \ell+1}$ close to 1 indicates that, after relabeling experts, most tokens follow consistent expert identities across the two layers.

### C.2.1. BACKWARD TRANSFER VIA CROSS-LAYER COUPLING

We now formalize the above intuition with a simple *backward-transfer* guarantee: if adjacent layers exhibit strong cross-layer coupling, then aligning the earlier-layer routing to the later-layer routing bounds how far it can drift from the same latent token allocation. For clarity, we restrict attention to the top-1 routing setting ($k = 1$), where each token activates exactly one expert per layer.

Specifically, let $\mathcal{T}$ denote the token space and let $t \sim D$ be drawn from the data distribution. Assume the existence of an underlying latent assignment

$$A : \mathcal{T} \to \{1, \ldots, E\},$$

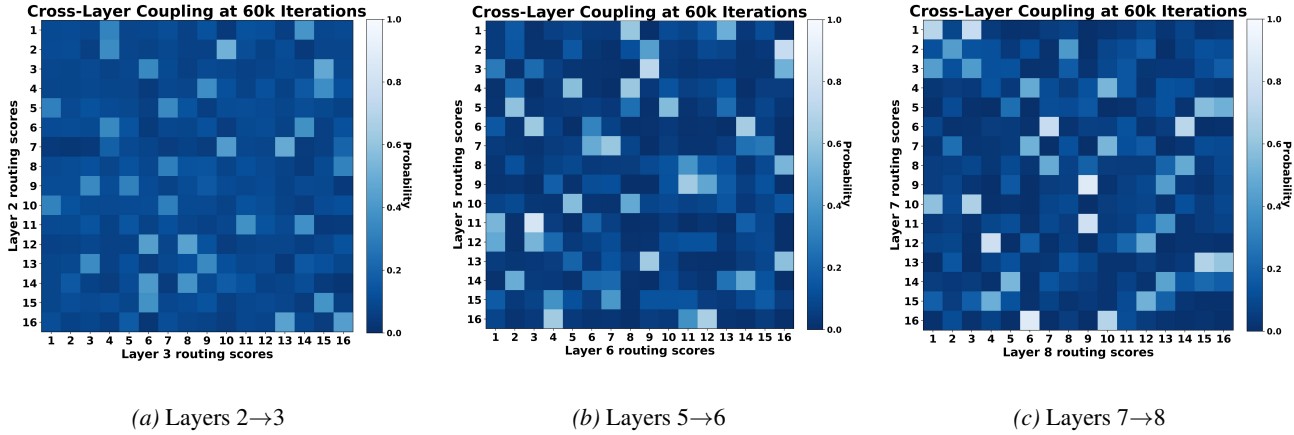

*(a)* Layers 2→3          *(b)* Layers 5→6          *(c)* Layers 7→8

*Figure 8.* Cross-layer coupling heatmaps. Coupling becomes more pronounced in deeper layers.

which represents an idealized data-induced expert partition.

For layer $\ell$, let $C_\ell(t) \in \{1, \ldots, E\}$ denote the expert selected by the router for token $t$. Define the permutation-invariant cross-layer coupling coefficient

$$\kappa_{\ell \to \ell+1} \;:=\; \max_{\pi \in S_E} \mathbb{P}_{t \sim D}\big[\pi\big(C_\ell(t)\big) = C_{\ell+1}(t)\big], \tag{42}$$

where $S_E$ is the symmetric group on $\{1, \ldots, E\}$.

Define the routing error of layer $\ell + 1$ relative to $A$ as

$$\varepsilon_{\ell+1} \;:=\; \mathbb{P}_{t \sim D}[C_{\ell+1}(t) \neq A(t)].$$

**Theorem C.5** (Backward transfer via cross-layer coupling)**.** *There exists a permutation $\pi^\star \in S_E$ such that the aligned routing decision $\pi^\star(C_\ell(t))$ satisfies*

$$\mathbb{P}_{t \sim D}\big[\pi^\star\big(C_\ell(t)\big) \neq A(t)\big] \;\leq\; \varepsilon_{\ell+1} + \big(1 - \kappa_{\ell \to \ell+1}\big). \tag{43}$$

*Proof.* By definition of $\kappa_{\ell \to \ell+1}$, there exists

$$\pi^\star \in \arg \max_{\pi \in S_E} \mathbb{P}_{t \sim D}\big[\pi\big(C_\ell(t)\big) = C_{\ell+1}(t)\big]$$

such that

$$\mathbb{P}_{t \sim D}\big[\pi^\star\big(C_\ell(t)\big) = C_{\ell+1}(t)\big] = \kappa_{\ell \to \ell+1}.$$

Consider the event

$$\mathcal{E} \;:=\; \big\{\pi^\star\big(C_\ell(t)\big) \neq A(t)\big\}.$$

Then

$$\mathcal{E} \subseteq \{C_{\ell+1}(t) \neq A(t)\} \;\cup\; \big\{\pi^\star\big(C_\ell(t)\big) \neq C_{\ell+1}(t)\big\}.$$

Taking probabilities and applying the union bound yields

$$\mathbb{P}(\mathcal{E}) \leq \mathbb{P}[C_{\ell+1}(t) \neq A(t)] + \mathbb{P}\big[\pi^\star\big(C_\ell(t)\big) \neq C_{\ell+1}(t)\big]$$

$$= \varepsilon_{\ell+1} + \Big(1 - \mathbb{P}\big[\pi^\star\big(C_\ell(t)\big) = C_{\ell+1}(t)\big]\Big)$$

$$= \varepsilon_{\ell+1} + \big(1 - \kappa_{\ell \to \ell+1}\big),$$

which proves the claim. □

Theorem C.5 explains why cross-layer agreement is a meaningful training signal. If layer $\ell+1$ routing is closer to an underlying token allocation (small $\kappa_{\ell+1}$), then forcing $g_\ell$ to align with $g_{\ell+1}$ bounds how far $g_\ell$ can drift from that same allocation. This gives a principled view of $\mathcal{R}_{\mathrm{cp}}$: later-layer routing can act as a "teacher signal" that transfers stable pathways backward, reducing routing ambiguity early in training.

## C.3. Compatibility with load balancing

In this section, we present the complete statements and proofs of Proposition C.6 and Proposition C.7 which regards the compatibility between the load balancing condition, the intra-layer specialization loss, and the cross-layer coupling loss.

Before presenting the proof, we note that *exact* load balancing is not achievable when the batch size is not divisible by the number of experts. However, since the imbalance per expert is at most one token, and the batch size in practice is large, this discrepancy is negligible. Thus, in this subsection, we assume the batch size is divisible by the number of experts without loss of generality.

### C.3.1. BALANCED PARTITION EXISTS UNDER ORTHOGONALITY-STYLE STRUCTURE

The following proposition demonstrates that load balancing can be maintained under conditions of expert orthogonality, illustrating the compatibility between the intra-layer specialization loss and load balancing:

**Proposition C.6.** *Suppose $k = 1$. For $e = 1, 2, \cdots, E$, denote $\mathcal{P}^{(\ell,e)}$ as the input space in which all the token can activate the $e$-th expert in layer $l$. Then there is always possible that the token space $\mathcal{P}^{(\ell,1)}, \mathcal{P}^{(\ell,2)}, \cdots, \mathcal{P}^{(\ell,E)}$ are convex, connected, and disjoint. Moreover, each $\mathcal{P}^{(\ell,e)}$ contains $B/E$ elements for the batch of input tokens.*

*Proof.* Since $E \mid B$, let $m = \dfrac{B}{E}$. We aim to partition the input set $\{x_1^{(\ell,e)}, x_2^{(\ell,e)}, \cdots, x_B^{(\ell,e)}\} \subset \mathbb{R}^h$ into $E$ convex connected subsets of equal size $m$. Pick any nonzero vector $a \in \mathbb{R}^h$. For each token $x_i^{(\ell,e)}$, compute the scalar projection

$$u_i = a^\top x_i^{(\ell,e)}, \quad i = 1, \ldots, N. \tag{44}$$

Without loss of generality, sort them in increasing order:

$$u_{(1)} \leq u_{(2)} \leq \cdots \leq u_{(N)}. \tag{45}$$

As $N_E \mid N$, we can divide this ordered list into $E$ consecutive blocks of size $m$. Specifically, denote

$$B_e := \{u_{((e-1)m+1)}, u_{((e-1)m+2)}, \ldots, u_{(em)}\}, \quad e = 1, \ldots, E. \tag{46}$$

Now define $E - 1$ hyperplanes of the form

$$H_e = \{x \in \mathbb{R}^n : a^\top x + b_r = 0\}, \qquad e = 1, \ldots, E - 1, \tag{47}$$

where each $b_e$ is chosen to satisfy that $-b_e \in \left(u_{(em)}, u_{(em+1)}\right)$, which means that the hyperplane lies strictly between the last element of block $B_r$ and the first element of block $B_{e+1}$.

These hyperplanes split $\mathbb{R}^h$ into $E$ slabs:

$$P_r = \{x : a^\top x \in [\alpha_{e-1}, \alpha_e]\}, \quad e = 1, \ldots, E, \tag{48}$$

where $\alpha_0 < \alpha_1 < \cdots < \alpha_E$ are thresholds satisfying

$$u_{(em)} < \alpha_e < u_{(em+1)}, \quad e = 1, \ldots, E - 1, \tag{49}$$

and we set $\alpha_0 = -\infty$, $\alpha_E = +\infty$ for completeness.

Each region $P_e$ is convex (intersection of halfspaces), connected, and by construction contains exactly $m = B/E$ tokens. Thus, if we let $\mathcal{P}^{(\ell,e)} = P_e$, we obtain a partition of the token space into $E$ disjoint convex connected subset with equal token counts, which proves the theorem. $\square$

Proposition C.6 is an existence result addressing a common concern: load balancing constrains *how much* each expert is used, while our regularizers constrain *what* experts learn. The proposition shows that even under strong orthogonality-style specialization, one can construct token partitions (of equal measure) and corresponding router directions that satisfy perfect load balance. Hence, balancing and specialization objectives need not be inherently in conflict.

### C.3.2. COUPLING OPTIMUM EXISTS UNDER PERFECT LOAD BALANCE

Then we consider the compatibility between the coupling loss. Formally, for a given input $x_i^{(\ell)}$ and expert $e = 1, 2, \cdots, E$, define a binary variable:

$$f_i^{(\ell,e)} := \chi(\text{The expert } e \text{ in layer } \ell \text{ is activated})$$

where $\chi$ denotes the indicator function. With the definition of $f_i^{(\ell,e)}$, we can present the following proposition:

**Proposition C.7.** *If we define the coupling loss $\mathcal{R}_{cp}(x_i)$ as Eq. (7), there exists a state that $\mathcal{L}_{cp}$ reach the optimal when satisifying the load balance condition*

$$\sum_{i=1}^{B} f_i^{(\ell,e)} = \sum_{i=1}^{B} f_i^{(\ell,\nu)}$$

*for any $e, \nu \in \{1, 2, \cdots, E\}$.*

*Proof.* Denote the coupling loss for one given token batch as $\mathcal{L}_{cp} := \sum_{i=1}^{B} \mathcal{R}_{cp}(x_i)$, then we have:

$$\mathcal{L}_{cp} = -\sum_{i=1}^{B}\sum_{\ell=1}^{L-1}\sum_{e=1}^{E}\sum_{\nu \in \mathbb{T}_i^{(\ell,e)}} s_i^{(\ell,e)} s_i^{(\ell+1,\nu)} \geq -\sum_{i=1}^{B}\sum_{\ell=1}^{L-1}\sum_{e=1}^{E} s_i^{(\ell,e)} \quad = -\left(B(L-1)\right), \tag{50}$$

where the equality condition is that for any $\nu \notin \mathbb{T}_i^{(\ell,e)}$ it holds

$$P_i^{(\ell,(e,\nu))} = 0, \tag{51}$$

for $i = 1, 2, \cdots, B$ and $e = 1, 2, \cdots, E$.

Recall the load balance condition

$$\sum_{i=1}^{B} f_i^{(\ell,e)} = \sum_{i=1}^{B} f_i^{(\ell,\nu)}, \tag{52}$$

where $e, \nu \in \{1, 2, \cdots, E\}$. We now prove that (51) and (52) can be simultaneously satisfied by explicitly constructing the desired condition.

We denote

$$[n] := \{1, 2, \ldots, n\}, \qquad [n]^k := \underbrace{[n] \times \cdots \times [n]}_{k \text{ times}}.$$

And we also define modular addition on $\{1, \ldots, n\}$ by

$$a \oplus_n b := \left((a - 1 + b) \bmod n\right) + 1.$$

Consider $\iota = (\iota_1, \ldots, \iota_k)$, any array in $[E]^k$. We define the following class of functions:

$$\mathcal{F}_{B,E,k} := \left\{ f : [B] \to [E]^k \;\middle|\; \forall i = 1, 2, \cdots, B, \; f(i) := \left(\iota_1 \oplus_{N_E} (i - 1), \ldots, \iota_k \oplus_{N_E} (i - 1)\right) \right\}.$$

Equivalently in component form, it holds that

$$f(i) = \left(f(i)_1, \ldots, f(i)_k\right), \quad (f(i))_r = \left((s_r - 1) + (i - 1)\right) \bmod N_E + 1, \quad r = 1, \ldots, k, \tag{53}$$

where $(f(i))_r$ denotes the $\mathcal{R}$-th element of $f(i)$. Then implies the recursion that

$$\forall i \in \{1, \ldots, B\}, \; \forall r \in \{1, \ldots, k\}, \quad f(i+1)_r = f(i)_r \oplus_{N_E} 1, \qquad f(N_E + 1) = f(1). \tag{54}$$

Now taking any collection of parameters $\eta_i^{(\ell,\kappa)}$ for $i = 1, 2, \cdots, B$, $\ell = 1, 2, \cdots, L$, and $\kappa = 1, 2, \cdots, k$ subject to the normalization constraint

$$\sum_{\kappa=1}^{k} \eta_i^{(\ell,\kappa)} = 1, \quad \forall \ell \in \{1, 2, \cdots, L\} \text{ and } i \in \{1, 2, \cdots, B\}. \tag{55}$$

We also take $f_1, f_2, \ldots, f_L \in \mathcal{F}_{B,E,k}$. Then define the routing scores $\eta_i^{(\ell,e)}$ by

$$s_i^{(\ell,e)} = \begin{cases} \eta_i^{(\ell,\kappa)}, & \text{if } e = (f_\ell(i))_\kappa \text{ for some } \kappa \in \{1,\ldots,k\}, \\ 0, & \text{otherwise.} \end{cases} \tag{56}$$

We now verify that the term $s_i^{(\ell,e)}$ defined in (56) satisfies conditions (51) and (52). Specifically, for Eq. (51) we have

$$P_i^{(\ell,(e,\nu))} := s_i^{(\ell,e)} s_i^{(\ell+1,\nu)} = \begin{cases} \eta_i^{(\ell,\kappa_1)} \eta_i^{(\ell+1,\kappa_2)}, & \text{if } e = (f_\ell(i))_{\kappa_1}, \ \nu = (f_{\ell+1}(i))_{\kappa_2}, \\ 0, & \text{otherwise.} \end{cases} \tag{57}$$

Thus $\mathbb{T}_i^{(\ell,e)}$ equals to the set of all the elements of $f_{\ell+1}(i)$. And for any $\nu \notin \mathbb{T}_i^{(\ell,e)}$, it holds that $P_i^{(\ell,(e,\nu))} = 0$.

Moreover, we consider Eq. (52). Recall the recursion property (54) of the selected function. Since $E|B$, in each layer every expert is loaded exactly $\dfrac{Bk}{R}$, which directly gives (52). $\qquad\square$

Proposition C.7 strengthens the compatibility message for $\mathcal{R}_{\mathrm{cp}}$: it shows that there exist routing configurations that minimize the coupling objective while maintaining perfect load balance. This supports treating $\mathcal{R}_{\mathrm{cp}}$ as a plug-and-play auxiliary term on top of standard balancing losses, rather than a competing constraint.

### C.4. Summary: closing the theoretical loop.

Appendix C can be read as answering three questions that together close the theory loop:

- **Why does specialization sharpen routing?** Appendix C.1 shows that even a weak best-expert advantage is sufficient to push routing toward a sharper, lower-entropy regime (Theorem C.1), making expert assignments more decisive.

- **Why does decisive routing amplify specialization?** Appendix C.1 further shows that once routing is sharp and stable, each expert is trained on higher-purity data, which strengthens experts on their advantage regions and amplifies specialization (Theorem C.4).

- **Why is cross-layer coupling a useful learning signal, and is it compatible with load balancing?** Appendix C.2 formalizes cross-layer coupling as a meaningful self-supervised signal: under strong coupling, later-layer routing can transfer pathway structure backward and stabilize token–expert identities across depth (Theorem C.5). Appendix C.3 establishes that these objectives remain compatible with standard load balancing (Propositions C.6 and C.7).

Together with the main-text propositions (proved in Appendix B), these results provide a coherent closed-loop theory for why $\mathcal{R}_{\mathrm{sp}}$ and $\mathcal{R}_{\mathrm{cp}}$ jointly improve both specialization quality and routing stability.

## D. Experimental details and additional experiments for pre-training tasks

### D.1. Experimental setup

**Infrastructure.** We integrate two auxiliary loss functions into the Megatron-LM framework (Shoeybi et al., 2019) as a plug-and-play module. By setting the corresponding hyperparameters, these losses can be enabled during MoE training.

**Model architecture.** We evaluate two MoE variants at multiple scales. For the vanilla MoE, we adopt a mainstream design comprising RMS normalization (Zhang & Sennrich, 2019), SwiGLU activations (Shazeer, 2020), and rotary position embeddings (RoPE) (Su et al., 2024); architectural hyperparameters are listed in Table 4. For the DeepSeek-style MoE, we augment the vanilla design with **ONE** shared expert and employ the **auxiliary-loss-free** load balancing strategy (Dai et al., 2024). The hyperparameters $\lambda_{\mathrm{cp}}$ and $\lambda_{\mathrm{sp}}$ are set to $1 \times 10^{-3}$ and $2 \times 10^{-3}$, respectively. Unless otherwise noted, the load-balance loss weight is set to $1 \times 10^{-2}$, the z-loss weight $\mathcal{R}_z$ (Zoph et al., 2022) to $1 \times 10^{-3}$, and the update step size for the coefficient $b$ in the auxiliary-loss-free load balancing strategy to $1 \times 10^{-3}$.

*Table 4.* Mixture-of-Experts (MoE) model configurations and training data volumes. 'A. Experts' denotes the activated experts and 'A. Params' denotes the activate parameters.

| Model size | Experts | A. Experts | Params | A. Params | Training Tokens |
|---|---|---|---|---|---|
| Small | 16 | 2 | 0.4B | 80M | 30B |
| Medium | 64 | 4 | 1.1B | 100M | 30B |
| Large | 96 | 6 | 7.0B | 500M | 50B |

*Table 5.* Ablations for two MoE architectures; metric is perplexity ($\downarrow$).

| Model | $\mathcal{L}_{lb}$ | $\mathcal{L}_{lb,sp}$ | $\mathcal{L}_{lb,cp}$ | $\mathcal{L}_{lb,sp,cp}$ | $\mathcal{L}_{lb,z}$ | $\mathcal{L}_{lb,z,sp}$ | $\mathcal{L}_{lb,z,cp}$ | $\mathcal{L}_{lb,z,sp,cp}$ |
|---|---|---|---|---|---|---|---|---|
| Vanilla MoE | 12.50 | 12.44 | 12.33 | 12.27 | 12.33 | 12.27 | 12.21 | 12.17 |
| DeepSeek-style MoE | 12.33 | 12.29 | 12.22 | 12.16 | 12.07 | 12.05 | 12.00 | 11.99 |

**Training settings.** Training is performed on the C4-en dataset (Raffel et al., 2020) using the LLaMA-2 tokenizer (Touvron et al., 2023). The small and medium MoE models are trained for 30 billion tokens, and the large MoE model for 50 billion tokens. This token budget exceeds the data size suggested by MoE scaling laws (Clark et al., 2022), providing sufficient signal for convergence. We use AdamW (Loshchilov & Hutter, 2017) optimizer with moment coefficient $\beta_1 = 0.9$, $\beta_2 = 0.999$, and weight decay coefficient 0.1.

### D.2. Ablation study for different regularizations

To evaluate the influence of various regularization techniques on model performance, we performed an ablation study utilizing a medium-scale architecture. The outcomes of this investigation are summarized in Table 5. Our analysis identifies several consistent trends. First, each auxiliary objective demonstrates individual efficacy: for the Vanilla MoE model, the introduction of $\mathcal{R}_{sp}$ leads to a reduction in perplexity, whereas $\mathcal{R}_{cp}$ produces a more substantial improvement. Similarly, in the DeepSeek-style MoE, both regularizers enhance performance, with $\mathcal{R}_{cp}$ yielding a greater effect. Moreover, the two losses exhibit complementarity, as their combined application results in further gains. When integrated with additional components such as $\mathcal{R}_{lb}$, the full regularization set achieves the most pronounced enhancements across both model variants. These patterns indicate that the specialization and coupling mechanisms independently contribute to refining expert behavior and, when employed together, synergize to produce cumulative reductions in perplexity.

### D.3. Downstream Task Evaluations for pre-trained MoE models.

We evaluate the pre-trained MoE models on supervised fine-tuning tasks (see Appendix for details; (Raffel et al., 2020)) and seven zero-shot benchmarks: BoolQ (Clark et al., 2019), ARC-Easy and ARC-Challenge (Clark et al., 2018), TruthfulQA-MC2 (Lin et al., 2022), PIQA (Bisk et al., 2020), MMLU (Hendrycks et al., 2020), and HellaSwag (Zellers et al., 2019) as outlined in Table 6. For each experimental setup, the process was conducted three times with different random seeds to ensure robustness.

Across both architectures, the addition of $\mathcal{R}_{cp}$ and $\mathcal{R}_{sp}$ enhances zero-shot accuracy in synergy with load-balance loss and z-loss. For the Vanilla MoE, integrating $\mathcal{R}_{sp}$ and $\mathcal{R}_{cp}$ with load-balance regularization leads to a marked improvement in average accuracy. Further gains are observed when $\mathcal{R}_{sp}$ and $\mathcal{R}_{cp}$ are applied in combination with load-balance loss and z-loss. In the DeepSeek-style MoE, a similar trend emerges: the inclusion of $\mathcal{R}_{sp}$ and $\mathcal{R}_{cp}$ alongside $\mathcal{R}_{lb}$ boosts average performance, while the loss with full set of regularization ($\mathcal{L}_{lb,z,sp,cp}$) achieves the highest overall accuracy, outperforming both $\mathcal{L}_{lb,z}$ and $\mathcal{L}_{lb}$, and yielding superior results on benchmarks such as ARC-E, ARC-C, and PIQA.

Although individual benchmarks show slight variations, the consistent upward trend in average accuracy demonstrates that $\mathcal{R}_{cp}$ and $\mathcal{R}_{sp}$ effectively complement load-balance loss and z-loss, contributing to downstream improvements across model families.

### D.4. The Impact of Auxiliary Loss on Load Balance Loss

To investigate the impact of the proposed regularization terms $\mathcal{R}_{sp}$ and $\mathcal{R}_{cp}$ on the primary training objective, we analyze the load balance loss curves throughout the training process. Figure 9 compares the load balance loss of the full model $\mathcal{R}_{lb,sp,cp}$

*Table 6.* Zero-shot accuracy of *Vanilla MoE* and *DeepSeek-style MoE* across seven benchmarks (↑).

| Model | Loss | BoolQ | ARC-E | ARC-C | Truthful QA-MC2 | PIQA | MMLU | Hella Swag | Avg. |
|---|---|---|---|---|---|---|---|---|---|
| Vanilla MoE | $\mathcal{L}_{\text{lb}}$ | $0.570_{\pm 0.003}$ | $0.452_{\pm 0.003}$ | $0.204_{\pm 0.003}$ | $0.432_{\pm 0.001}$ | $0.622_{\pm 0.005}$ | $0.247_{\pm 0.002}$ | $0.268_{\pm 0.002}$ | 0.399 |
| | $\mathcal{L}_{\text{lb,sp,cp}}$ | $0.578_{\pm 0.003}$ | $0.462_{\pm 0.002}$ | $0.210_{\pm 0.004}$ | $0.451_{\pm 0.003}$ | $0.627_{\pm 0.002}$ | $0.253_{\pm 0.002}$ | $0.275_{\pm 0.004}$ | **0.408** |
| | $\mathcal{L}_{\text{lb,z}}$ | $0.567_{\pm 0.003}$ | $0.457_{\pm 0.004}$ | $0.205_{\pm 0.002}$ | $0.433_{\pm 0.003}$ | $0.629_{\pm 0.002}$ | $0.250_{\pm 0.001}$ | $0.267_{\pm 0.004}$ | 0.401 |
| | $\mathcal{L}_{\text{lb,z,sp,cp}}$ | $0.589_{\pm 0.003}$ | $0.453_{\pm 0.004}$ | $0.206_{\pm 0.006}$ | $0.445_{\pm 0.003}$ | $0.637_{\pm 0.003}$ | $0.257_{\pm 0.002}$ | $0.274_{\pm 0.003}$ | **0.409** |
| DS-style MoE | $\mathcal{L}_{\text{lb}}$ | $0.578_{\pm 0.002}$ | $0.453_{\pm 0.001}$ | $0.205_{\pm 0.003}$ | $0.438_{\pm 0.003}$ | $0.631_{\pm 0.002}$ | $0.248_{\pm 0.001}$ | $0.269_{\pm 0.002}$ | 0.403 |
| | $\mathcal{L}_{\text{lb,sp,cp}}$ | $0.584_{\pm 0.001}$ | $0.452_{\pm 0.003}$ | $0.206_{\pm 0.005}$ | $0.457_{\pm 0.002}$ | $0.635_{\pm 0.002}$ | $0.255_{\pm 0.003}$ | $0.277_{\pm 0.005}$ | **0.410** |
| | $\mathcal{L}_{\text{lb,z}}$ | $0.564_{\pm 0.002}$ | $0.453_{\pm 0.002}$ | $0.205_{\pm 0.002}$ | $0.444_{\pm 0.002}$ | $0.628_{\pm 0.001}$ | $0.252_{\pm 0.001}$ | $0.270_{\pm 0.004}$ | 0.402 |
| | $\mathcal{L}_{\text{lb,z,sp,cp}}$ | $0.575_{\pm 0.002}$ | $0.461_{\pm 0.004}$ | $0.214_{\pm 0.004}$ | $0.452_{\pm 0.004}$ | $0.642_{\pm 0.003}$ | $0.257_{\pm 0.002}$ | $0.280_{\pm 0.002}$ | **0.412** |

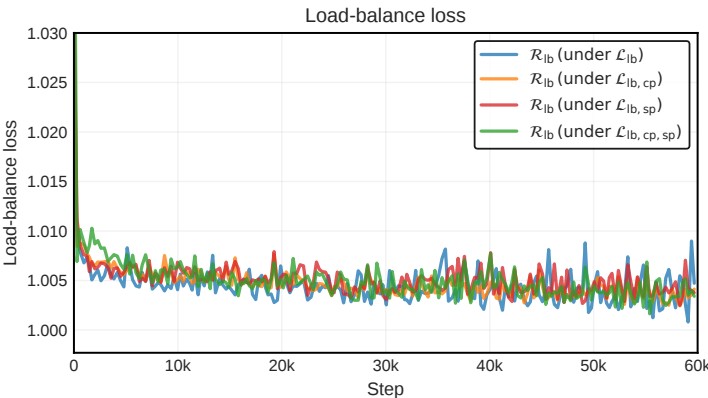

*Figure 9.* The impact of auxiliary loss on load balance loss

with ablation studies involving the baseline configurations $\mathcal{R}_{\text{lb}}$, $\mathcal{R}_{\text{lb,cp}}$, and $\mathcal{R}_{\text{lb,sp}}$.

All model variants exhibit rapid and stable convergence. The load balance loss for each configuration declines sharply within the initial 5,000 steps and stabilizes promptly near its optimal value. This demonstrates that incorporating the auxiliary objectives does not hinder the model's capacity to learn the primary load balancing task.

The load-balance loss curves are nearly identical across all configurations, with the trajectories largely overlapping throughout training. This indicates that incorporating $\mathcal{R}_{\text{sp}}$ and $\mathcal{R}_{\text{cp}}$ has a negligible effect on load balancing and does not alter the optimization of the load-balance objective.

### D.5. Pre-training results with random seeds

To rigorously demonstrate that the reported improvements are attributable to the auxiliary regularization and are statistically significant rather than resulting from optimization noise, we conducted repeated pre-training experiments using medium-sized models for both the Vanilla MoE and DeepSeek-style architectures. The experimental configuration remains identical to that described in Appendix D.1.

As illustrated in the Table 7, for the Vanilla MoE, the comparison between $\mathcal{L}_{\text{lb}}$ versus $\mathcal{L}_{\text{lb,sp,cp}}$ shows an improvement from 12.50 to 12.26, corresponding to an approximately 1.9% relative reduction, with a standard deviation across seeds of only 0.01. Furthermore, when all auxiliary terms are included, $\mathcal{L}_{\text{lb,z}}$ versus $\mathcal{L}_{\text{lb,z,sp,cp}}$ improves from 12.33 to 12.17 , with standard deviations ranging from 0.01 to 0.02. These findings confirm consistent and statistically meaningful gains in validation performance.

### D.6. Hyperparameter sensitivity in pre-training

The validation performance for the pre-training tasks, as presented in Table 1, is based on a fixed hyperparameter selection described in Appendix D.1. To examine the sensitivity of the hyperparameters $\lambda_{\text{cp}}$ and $\lambda_{\text{sp}}$, we performed a hyperparameter

*Table 7.* Validation perplexity for medium model scale with three random repetitions ($\downarrow$).

| Losses | Vanilla MoE | DeepSeek-style MoE |
|---|---|---|
| $\mathcal{L}_{\text{lb}}$ | 12.50 (0.01) | 12.33 (0.02) |
| $\mathcal{L}_{\text{lb,sp,cp}}$ | 12.26 (0.01) | 12.15 (0.02) |
| $\mathcal{L}_{\text{lb,z}}$ | 12.33 (0.02) | 12.07 (0.01) |
| $\mathcal{L}_{\text{lb,z,sp,cp}}$ | 12.17 (0.01) | 11.98 (0.01) |

*Table 8.* Perplexity with fixed $\lambda_{\text{sp}}$ and varied $\lambda_{\text{cp}}$.

| $\lambda_{\text{cp}}$ | $2 \times 10^{-4}$ | $5 \times 10^{-4}$ | $1 \times 10^{-3}$ | $2 \times 10^{-3}$ |
|---|---|---|---|---|
| PPL | 12.41 | 12.35 | **12.27** | 12.30 |

*Table 9.* Perplexity with fixed $\lambda_{\text{cp}}$ and varied $\lambda_{\text{sp}}$.

| $\lambda_{\text{sp}}$ | $5 \times 10^{-4}$ | $1 \times 10^{-3}$ | $2 \times 10^{-3}$ | $3 \times 10^{-3}$ |
|---|---|---|---|---|
| PPL | 12.32 | 12.30 | **12.27** | 12.29 |

sweep around the default values using a medium-sized model. Validation perplexity (where lower values indicate better performance) was measured under the following variations:

- With $\lambda_{\text{sp}}$ fixed at $2 \times 10^{-3}$, we varied $\lambda_{\text{cp}}$ from 0.2 to 2 times the default value of $1 \times 10^{-3}$.

- With $\lambda_{\text{cp}}$ fixed at $1 \times 10^{-3}$, we varied $\lambda_{\text{sp}}$ from 0.25 to 3 times the default value of $2 \times 10^{-3}$.

The results, shown in Tables 8 and 9, demonstrate that the model performance remains stable across a broad interval. The perplexity changes are limited to less than 1% relative to the optimum. The heuristic choice of $\lambda_{\text{cp}} = 10^{-3}$ and $\lambda_{\text{sp}} = 2 \times 10^{-3}$ yields near-optimal results, and deviations cause only minor degradation.

### D.7. Quantitative comparison with DeepSeekMoE-style load balancing

To directly address whether training with our proposed specialization induces more expert specialization than DeepSeek-MoE's auxiliary-loss-free load balancing, we compare two training objectives including $\mathcal{L}_{\text{lb}}$ and $\mathcal{L}_{\text{lb,cp,sp}}$ over the small model with configurations in Table 4. As a proxy for expert specialization and routing coherence, we measure every 1000 iterations the percentage of tokens whose top-1 expert assignment remains unchanged between consecutive checkpoints. Higher values correspond to more stable token–expert assignments, lower routing entropy, and, via Proposition C.7, more persistent expert-specific gradient directions.

The results, as detailed in Table 10, demonstrate that across all training stages, $\mathcal{L}_{\text{lb,cp,sp}}$ consistently enhances routing stability by 1–2 absolute points (approximately 2%–4% relative improvement) compared to $\mathcal{L}_{\text{lb}}$. The gains are especially significant during early training phases when routing ambiguity is most severe, which aligns precisely with the regime addressed by Proposition 2. Furthermore, the benefits persist even at later stages (e.g., 59K–60K iterations), indicating that expert assignments maintain greater consistency over time.

In conjunction with Fig 3, where $\mathcal{L}_{\text{lb,sp,cp}}$ reduces the intra-layer specialization loss $R_{\text{sp}}$ relative to $\mathcal{L}_{\text{lb,sp}}$, these findings confirm that our method further sharpens expert differentiation beyond the capabilities of auxiliary-loss-free load balancing alone. This improvement is consistent with our theoretical framework: reduced activation similarity promotes more orthogonal gradients (as per Proposition C.7), while enhanced routing stability supports stronger and more coherent expert paths (in line with Proposition 5.1).

## E. Finetuning tasks evaluation

We fine-tune `Qwen3-30B-A3B-Instruct-2507` model on an internal corpus of 38B tokens under identical training hyperparameters listed in Table 11. We evaluate on a broad suite of reasoning and knowledge-intensive benchmarks, including HumanEval (Chen et al., 2021), GSM8K (Cobbe et al., 2021), math500_PRM800K_dataset (Lightman et al., 2023), and MMLU (Hendrycks et al., 2020). Across nearly all settings, incorporating $\mathcal{R}_{\text{cp}}$ and $\mathcal{R}_{\text{sp}}$ outperforms the baseline, yielding consistent gains on reasoning-oriented tasks as well as aggregate knowledge measures as Table 12. While a minor fluctuation is observed on the humanities subset of MMLU, the overall trend remains positive, confirming that our objectives not only sharpen specialization in pre-training but also transfer effectively to finetuning adaptation.

*Table 10.* The fraction of tokens that keep the same experts between checkpoints.

| Iteration range | 1K–2K | 2K–3K | 4K–5K | 9K–10K | 19K–20K | 29K–30K | 59K–60K |
|---|---|---|---|---|---|---|---|
| $\mathcal{L}_{\text{lb,sp}}$ | 0.4746 | 0.6056 | 0.6601 | 0.6987 | 0.7450 | 0.7864 | 0.9011 |
| $\mathcal{L}_{\text{lb,sp,cp}}$ | 0.4898 | 0.6213 | 0.6757 | 0.7187 | 0.7594 | 0.7935 | 0.9067 |

*Table 11.* Hyperparameters for the fine-tuning task under Qwen3-30B models.

| Hyperparameters | Value |
|---|---|
| Global batch size | 64 |
| Learning rate | 8e-5 |
| Epochs | 3 |
| Sequence length | 32768 |
| $\lambda_{\text{lb}}^{\diamond}$ | 1e-3 |
| $\lambda_{\text{sp}}$ | 2e-4 |
| $\lambda_{\text{cp}}$ | 1e-4 |

$^{\diamond}$ The coefficient of the regularization of load-balancing.

## F. Inference acceleration

To leverage the benefits of the specialization loss and coupling loss during the inference period, we implement a path-aware placement and bucketing strategy. This involves estimating a cross-layer expert co-activation matrix from a held-out dataset, greedily co-locating strongly coupled experts on the same GPU shard via graph partitioning, and performing a lightweight pre-routing pass through the first MoE router to bucket sequences according to early expert decisions. These buckets are then assigned to shards hosting the corresponding experts, ensuring that most subsequent dispatches remain local.

We evaluate our approach on MoE models of varying scales under 8 Nvidia A100 80G GPUs with expert parallelism. The number of parallel devices are set to 8 and the microbatch size is set to 1. A baseline model trained solely with $\mathcal{R}_{\text{lb}}$ is compared against our variant trained with $\mathcal{R}_{\text{lb,sp,cp}}$. While the baseline uses default round-robin expert placement and uniform batching, our model employs the path-aware scheme described above. We also apply identical system-level optimizations to both the load-balancing baseline and our model. This design cleanly separates the acceleration attributable to engineering infrastructure from that enabled by structural properties—specifically, stronger cross-layer expert coupling and lower routing entropy—induced by our proposed losses.

As summarized in Table 13, throughput improves consistently across model sizes and benchmarks—without any architectural modifications or additional parameters. These results demonstrate that reducing routing ambiguity through $\mathcal{R}_{\text{sp}}$ and $\mathcal{R}_{\text{cp}}$ directly enhances system-level efficiency by streamlining token-to-expert execution paths. With the inference throughput, it can be observed that our proposed auxiliary losses improve model perplexity while simultaneously enhancing inference efficiency through reduced routing entropy. By promoting sharper expert specialization and stronger cross-layer coupling, tokens follow more consistent expert paths, which in an expert parallelism setup improves cache locality and reduces All-to-All communication overhead.

## G. Comparison with recent auxiliary-loss methods for specialization

Several recent studies have introduced auxiliary loss functions aimed at improving expert specialization and routing efficacy in Mixture-of-Experts (MoE) models. In this section, we present a conceptual analysis and empirical evaluation comparing our approach with a representative method by (Guo et al., 2025a), which combines an orthogonality loss with a *variance* loss applied to the routing logits.

### G.1. Conceptual Comparison

Our method integrates two complementary components: the *intra-layer specialization* loss $\mathcal{L}_{\text{sp}}$, which promotes orthogonality in the representations of co-activated experts, thereby aligning their parameter gradients along orthogonal directions (see Proposition 4.1), and the *cross-layer coupling* loss $\mathcal{L}_{\text{cp}}$, which enforces consistency in expert selection across adjacent layers,

*Table 12.* Evaluation score on Qwen3-30B-A3B-Instruct-2507 finetuning tasks. The last four rows stands for the performance for mmlu dataset with different domains.

| Dataset | Metric | $\mathcal{L}_{lb}$ | $\mathcal{L}_{lb,sp,cp}$ |
|---|---|---|---|
| openai_humaneval | humaneval_pass@1 | 92.07 | **95.73** |
| gsm8k | accuracy | 93.33 | **94.16** |
| math_prm800k_500 | accuracy | 94.00 | **94.20** |
| mmlu | naive_average | 78.97 | **79.86** |
| mmlu-weighted | weighted_average | 76.35 | **77.10** |
| mmlu-humanities | naive_average | **75.90** | 75.42 |
| mmlu-stem | naive_average | 87.38 | **88.97** |
| mmlu-social-science | naive_average | 75.91 | **77.11** |
| mmlu-other | naive_average | 72.59 | **73.52** |

*Table 13.* Throughput comparison (samples/s; ↑) on four standard benchmarks. Here 'SO' means the system optimization.

| Model size | Loss | MMLU | GSM8K | HumanEval | Math500 |
|---|---|---|---|---|---|
| Small | $\mathcal{L}_{lb}$ *w.o.* SO | 161.3 (1.00×) | 26.2 (1.00×) | 35.7 (1.00×) | 6.9 (1.00×) |
| | $\mathcal{L}_{lb}$ *w.* SO | 164.9 (1.03×) | 26.5 (1.01×) | 36.6 (1.02×) | 7.0 (1.01×) |
| | $\mathcal{L}_{lb,sp,cp}$ *w.* SO | 170.4 (1.06×) | 27.0 (1.03×) | 37.5 (1.05×) | 7.1 (1.03×) |
| Medium | $\mathcal{L}_{lb}$ *w.o.* SO | 157.4 (1.00×) | 25.9 (1.00×) | 27.7 (1.00×) | 6.1 (1.00×) |
| | $\mathcal{L}_{lb}$ *w.* SO | 162.7 (1.03×) | 26.2 (1.01×) | 28.6 (1.03×) | 6.2 (1.01×) |
| | $\mathcal{L}_{lb,sp,cp}$ *w.* SO | 165.7 (1.05×) | 26.6 (1.03×) | 29.4 (1.06×) | 6.3 (1.03×) |
| Large | $\mathcal{L}_{lb}$ *w.o.* SO | 96.9 (1.00×) | 15.0 (1.00×) | 12.8 (1.00×) | 3.9 (1.00×) |
| | $\mathcal{L}_{lb}$ *w.* SO | 96.9 (1.00×) | 15.0 (1.00×) | 12.8 (1.00×) | 3.9 (1.00×) |
| | $\mathcal{L}_{lb,sp,cp}$ *w.* SO | 103.5 (1.07×) | 15.7 (1.05×) | 13.7 (1.07×) | 4.2 (1.08×) |

reducing routing ambiguity (see Proposition 5.1). These losses operate on principles of *information geometry and path consistency* and are compatible with standard router-stabilization techniques, such as the $z$-loss and logit clipping.

In contrast, the approach by (Guo et al., 2025a) incorporates an orthogonality term along with a *variance-maximization* objective on the routing logits, explicitly encouraging high logit dispersion to enhance discrimination. While increased dispersion can sharpen top-$k$ selections, it lacks inherent control over logit magnitudes, potentially leading to adverse interactions with softmax temperature and $z$-loss penalties. Specifically, unregulated variance amplification often causes prematurely peaked routing distributions or numerical instabilities (e.g., gradient spikes), increasing sensitivity to learning rate and initialization in large-scale pre-training. Our design mitigates these issues by regularizing *activations and paths* rather than directly inflating raw logit variance.

### G.2. Experimental Evaluation

We compare our losses with $\mathcal{L}_{lb,o,v}$ in Table 1 and Table 2, we show that our consistency-based formulation demonstrates stronger accuracy and stability. The combination of $\mathcal{L}_{sp}$ and $\mathcal{L}_{cp}$ outperforms variance-maximization methods in both pre-training and downstream settings. These gains align with our theoretical framework: (i) orthogonalizing expert activations yields orthogonal gradient directions, reducing parameter interference; (ii) cross-layer coupling concentrates routing probability mass along consistent paths, diminishing ambiguity and consolidating expert specialization. Together, these effects enhance final model quality and training dynamics for large-scale applications.

## H. Analysis for the computational and memory efficiency

In this section, we present a series of analysis for the computation and memory overhead for the gradient evaluation with our proposed auxiliary regularization.

*Table 14.* The iteration time and peak memory with different loss

| Model size | Loss | Iteration time (ms/iteration) | Peak memory (GB) |
|---|---|---|---|
| Small | $\mathcal{L}_{\text{lb}}$ | 405.9 (1.0000×) | 43.5 (1.0000×) |
| | $\mathcal{L}_{\text{lb,sp,cp}}$ | 413.6 (1.0190×) | 43.6 (1.0023×) |
| Medium | $\mathcal{L}_{\text{lb}}$ | 518.4 (1.0000×) | 60.6 (1.0000×) |
| | $\mathcal{L}_{\text{lb,sp,cp}}$ | 526.5 (1.0156×) | 60.7 (1.0016×) |
| Large | $\mathcal{L}_{\text{lb}}$ | 2927.8 (1.0000×) | 73.1 (1.0000×) |
| | $\mathcal{L}_{\text{lb,sp,cp}}$ | 2942.4 (1.0049×) | 73.3 (1.0027×) |

## H.1. Theoretical analysis for computational and memory overhead

Here we present a theoretical analysis for computational and memory overhead. From a computational perspective, both losses are lightweight relative to the model's core operations (attention mechanisms and feed-forward networks)

### H.1.1. INTRA-LAYER SPECIALIZATION LOSS ($\mathcal{R}_{\text{SP}}$).

**Computational Complexity**: This loss requires computing pairwise cosine similarities between activations of the top-k selected experts. For a hidden dimension $d$ and $k$ activated experts, the per-token complexity is $\mathcal{O}(k^2 \cdot d)$. In standard MoE configurations, $k$ typically assumes small values (e.g., 2 or 4), while $d$ represents a large dimension (e.g., 4096). Consequently, $k^2 \ll d$, rendering the cost of $\mathcal{O}(k^2 \cdot d)$ negligible compared to the standard FFN transformation cost of $\mathcal{O}(k \cdot d^2)$.

**Scalability**: Crucially, this computational cost depends solely on the number of activated experts $k$, rather than the total number of experts $E$. This implies that even as the total expert count $E$ scales to hundreds or thousands (as in "Mixture of Million Experts" architectures), as long as the activated expert count $k$ remains small, the computational overhead of $\mathcal{R}_{sp}$ remains both constant and minimal.

**Memory Requirements**: No additional memory allocation is necessary, as this loss reuses intermediate activations $z^{(l,e)}$ already computed during the forward pass.

### H.1.2. CROSS-LAYER COUPLING LOSS ($\mathcal{R}_{\text{CP}}$).

**Computational Complexity**: This loss operates exclusively on scalar routing logits. Specifically, it involves basic statistical operations on token routing scores across consecutive layers. These operations avoid complex matrix computations involving high-dimensional hidden states.

**Memory Requirements**: This loss requires storing a lightweight tensor of dimensions $E \times E \times L$ to track expert transition statistics. Since the number of experts $E$ is typically much smaller than the hidden dimension $d$, the memory consumption of this tensor is negligible.

## H.2. Empirical wall-clocked time and memory analysis

Empirical results are fully consistent with our theoretical complexity analysis. We systematically measured training throughput (in ms/iteration) and peak GPU memory consumption (in GB) across the Small (0.4B), Medium (1.1B), and Large (7.0B) model configurations used in Appendix D.1, with all experiments conducted on a uniform hardware configuration consisting of 8 A100 GPUs.

Our benchmarking results, as shown in Table 14, demonstrate that the overhead introduced by $\mathcal{R}_{\text{sp}}$ and $\mathcal{R}_{\text{cp}}$ is negligible: the combined auxiliary losses introduce only $0.5\%$ to $1.9\%$ additional latency, with the relative overhead exhibiting a decreasing trend as model scale increases (reducing to approximately $0.5\%$ for the 7B parameter model), indicating favorable scaling characteristics of our method, while the additional memory footprint is minimal ($< 0.3\%$), empirically confirming that our approach does not impose additional hardware requirements.

