# OpenReview forum: "Synergistic Intra- and Cross-Layer Regularization Losses for MoE Expert Specialization"
_ICML.cc/2026/Conference — ICML 2026 regular_

### Official Review · Reviewer_i9LV · 2026-03-04

**Soundness:** 3
**Presentation:** 3
**Significance:** 2
**Originality:** 2
**Overall Recommendation:** 4
**Confidence:** 3

**Summary:**

This paper addresses the expert overlap issue of MoE that results in redundant representations across experts and routing ambiguity, which wastes model capacity. Compared to some existing methods that address this issue by architectural design, the paper introduces two regularization losses that enhance MoE specialization and routing efficiency, which are compatible with existing methods. The first
intralayer specialization loss penalizes cosine similarity between experts’ SwiGLU activations on identical tokens, encouraging experts to specialize in complementary knowledge. And the cross-layer coupling loss maximizes joint Top-k routing probabilities across adjacent layers, aims at coherent expert pathways through network depth while reinforcing intra-layer expert specialization.

**Compliance With Llm Reviewing Policy:**

Affirmed.

**Key Questions For Authors:**

-The proposed losses are orthogonal to existing methods. I am curious to what extent the existing structural modification-based solutions are still needed after applying the losses to MoE, or whether they are complementary to each other?

-Some discussion about the limitations of methods.

**Limitations:**

Limitations are not discussed in the paper; it would be good to know in what situations the two losses give negative or small effects (say, the models themselves already contain structural improvements).

**Strengths And Weaknesses:**

Strengths

-The proposed losses are plug and play, so they do not need to alter model architectures that make them more deployable to MOE model. And they are orthogonal to the standard load-balancing loss and compatible with both the shared-expert architecture in DeepSeekMoE and vanilla top-k MoE architectures.

-Theoretical analyses are conducted to support claims of the effectiveness of two losses.

-comprehensive experiments for evaluating the effects of two losses.

-The organization of the paper is clear, the motivation, objectives, and methods are clearly mentioned.

Weaknesses:

-The method novelty of the two losses is not obvious, although the effects are shown in the experiments.

-The limitations are not discussed by using these two losses. In what situations they fail to improve, it is good to know when to apply them.

---

> ### Author Rebuttal · Authors · 2026-03-31
>
> ### 1. On novelty of the two losses (W1)
>
> Thank you for this important question. We agree that the individual algebraic forms of the losses are simple, and our claim is not that they are architecturally complex by themselves. Rather, the novelty lies in the objective design and the mechanism it targets: we treat expert specialization as a first-class training objective instead of relying on router/architecture redesign. Concretely, $R\_{sp}$ regularizes co-activated experts on the same token in activation space, directly reducing functional overlap, while $R\_{cp}$ turns the empirically observed cross-layer pathway continuity into an explicit supervision signal. To our knowledge, this combination—intra-layer anti-redundancy + cross-layer pathway coupling as plug-and-play objectives—has not been studied in this way. The contribution is further strengthened by the theory: the paper does not present the two losses as isolated heuristics, but connects them through a specialization–routing feedback loop, showing why they are synergistic rather than ad hoc.
>
> ### 2. On limitations and when the gains may be small (W2, L1)
> Thank you for this important question. We agree that the paper should discuss this more explicitly, and we will add a dedicated limitations paragraph in the revision. Based on our analysis and experiments, the method may help less in at least the following regimes. First, $R\_{cp}$ is most meaningful when adjacent MoE layers process related representations; if an architecture induces very large representation shifts between adjacent layers, the coupling signal may become weaker or less informative. Second, if future architectures allow tokens to skip MoE layers entirely (e.g., token-level depth routing), then the current adjacent-layer formulation of $R\_{cp}$ would need to be adapted to the actual traversed MoE path. Third, when a model already contains strong built-in specialization priors, the incremental gain from our auxiliary losses may naturally become smaller, even if still positive. Finally, as with most auxiliary objectives, overly large coefficients could in principle over-sharpen routing and reduce flexibility; however, in the tested range we did not observe consistent degradation, and the hyperparameter sweep shows a fairly broad stable region.
>
> ### 3. Complementarity with Structural MoE Methods (Q1)
> Thank you for this valuable suggestion. We view the two directions as complementary rather than mutually exclusive. Structural methods change the architecture or router design (e.g., shared experts, expert granularity, routing mechanisms), whereas our method changes only the training objective and can be added without modifying model code paths. In that sense, our losses are lower-cost and more portable, while structural methods encode stronger inductive biases. Empirically, our results already support this complementarity: both Vanilla MoE and DeepSeek-style MoE benefit from $R\_{sp}$ and $R\_{cp}$ (Table 1 / Table 5), which shows that the losses still help even when structural specialization priors already exist. At the same time, in some settings the improved Vanilla MoE approaches or even surpasses a stronger structural baseline, suggesting that targeted objectives can sometimes recover a substantial fraction of the gains of architectural redesign. So our position is: structural methods are still useful, especially when one wants stronger capacity allocation priors, but our losses remain valuable because they are architecture-agnostic, easy to integrate, and additive to those methods.
>
> ### 4. On possible negative effects / failure cases (Q2, L1)
> Thank you for this valuable suggestion. In our current experiments, we did not observe a systematic negative effect within the tested coefficient range; the gains are generally consistent across scales and architectures. That said, we agree it is important to be explicit that our method is not expected to help equally in every regime. Our current evidence suggests that the most likely “small-gain” cases are not catastrophic failures, but rather settings where the architecture already enforces specialization strongly, or where adjacent-layer coupling is inherently less meaningful. We will make this boundary clearer in the revised paper so that readers understand when these losses are most likely to be useful.

---

> > ### Author Rebuttal · Reviewer_i9LV · 2026-04-03
> >
> > My concerns have been addressed.

---

> > > ### Author Response · Authors · 2026-04-03
> > >
> > > Dear Reviewer i9LV,
> > >
> > > Thank you for your careful review and for your positive feedback. We are very glad that our response has fully clarified your concerns.
> > >
> > > Best regards,
> > >
> > > Authors of Submission 464

---

### Official Review · Reviewer_3n1a · 2026-03-09

**Soundness:** 2
**Presentation:** 3
**Significance:** 2
**Originality:** 3
**Overall Recommendation:** 4
**Confidence:** 4

**Summary:**

This paper addresses the issues of expert overlap and routing ambiguity in sparse Mixture-of-Experts (MoE) models. The authors propose two plug-and-play regularization losses: an intra-layer specialization loss that penalizes cosine similarity between expert activations, and a cross-layer coupling loss that encourages stable expert pathways across adjacent layers. Theoretical analysis links activation orthogonality to gradient decorrelation suggesting that reducing activation similarity may improve expert specialization, while experiments across various scales (up to 7B) show improvements in perplexity and downstream tasks.

**Compliance With Llm Reviewing Policy:**

Affirmed.

**Final Justification:**

My concerns have been addressed.

**Key Questions For Authors:**

1. In large-scale multi-node training using Expert Parallelism, the Top-k experts for a single token are likely distributed across different nodes. Does the proposed methods necessitate additional P2P or All-to-All communication to synchronize expert activations? Please clarify the potential communication bottleneck in a multi-node setup beyond the 8-GPU environment tested.
2. Could the authors provide visualizations to prove that the "coherent pathways" correspond to distinct functional or semantic domains.
3. The cross-layer coupling forces tokens into stable pathways, which may hinder the model's ability to dynamically combine knowledge for complex, cross-domain reasoning. How does this method perform on tasks requiring high routing flexibility or long-range context dependencies?
4. Does the cross-layer coupling loss risk prematurely locking tokens into suboptimal routing patterns during early training? Have the authors observed any cases where strong coupling leads to reduced routing diversity or degraded performance?

**Limitations:**

No.
The authors should discuss whether enforcing stable “coherent pathways” may reduce routing diversity, which is one of the core advantages of MoE architectures.
Additionally, the scalability of the proposed losses in large-scale distributed training should be discussed more explicitly.

**Strengths And Weaknesses:**

Strengths
1. The paper provides a clear mathematical link between activation cosine similarity and gradient alignment, providing a clear motivation.
2. The method is architecture-agnostic and does not modify the router or model structure, making it easy to integrate into existing MoE training pipelines.
3. The implementation is lightweight, with negligible training overhead(~0.5% latency for 7B models).
4. The authors include ablation studies, sensitivity analysis, and multi-seed repetitions to support their claims.

Weaknesses
1. The perplexity (PPL) gains in Table 1 are marginal (~2%) and fall within the range of hyperparameter sensitivity. This minimal pre-training gain does not align well with the significantly larger improvements reported in Tables 2 and 3, raising questions about the robustness of the downstream results.
2. While expert specialization is a known challenge, the proposed dual-loss framework increases system complexity. Although the cosine similarity computation itself is lightweight, computing pairwise similarity between expert activations may require gathering activations across devices when the top-k experts for a token reside on different nodes in expert-parallel training. The potential communication overhead is not discussed.
3. The "coherent pathways" enforced by $R_{cp}$ may suppress the inherent advantage of dynamic routing. There is a lack of evidence that such coupling remains beneficial for complex, cross-domain content where routing flexibility is typically essential.
4. The paper lacks a deep qualitative analysis of expert specialization (e.g., token-to-expert semantic mapping) and expert pathway evolution to definitively prove the mechanism behind the observed gains.

---

> ### Author Rebuttal · Authors · 2026-03-31
>
> ### 1. Why Small PPL Gains Can Still Matter (W1)
> The $\sim 2\\%$ PPL gain is not tuning noise: it is reproduced across 3 random seeds, appears consistently on both Vanilla and DeepSeek-style MoE, and remains stable across a wide hyperparameter range. The ablations also show that $\mathcal{R}\_{sp}$-only and $\mathcal{R}\_{cp}$-only each help, with the combination performing best.
> Beyond absolute PPL, the gains also appear in scaling efficiency: In Figure 5 and Figure 6, activating only $N=6$ experts outperforms the baseline with $N=10$ (PPL 12.13 vs. 12.15), and using fewer total experts of E=96 surpasses the baseline with $E=112$ (PPL 11.93 vs. 11.94). In other words, the same or better quality is reached with fewer active experts or a smaller expert pool.
>
> ### 2. Communication Overhead Under Expert Parallelism (W2, Q1)
> We thank the reviewer for this important point. We agree that in expert-parallel (EP) settings, if co-activated experts for the same token reside on different EP ranks, specialization loss may introduce extra communication and GPU memory overhead. However, with proper systems design, this overhead can be largely reduced: the cosine-similarity computation can be decoupled from the main forward pass and executed asynchronously on a separate communication stream, overlapping with communication-free computation rather than the critical-path MoE all-to-all; meanwhile, memory overhead can be bounded by using a fixed-size reusable buffer that computes similarities chunk by chunk, so it depends on the buffer size rather than the total number of tokens in the batch.
> To make this concrete, we additionally evaluate the overhead in a multi-node large-batch setting: Qwen3-30B-A3B pre-training on 4 nodes / 32 H20 GPUs (96GB), with EP$=32$, sequence length $=4096$, and batch size $=2048$. The measured throughput is 33868.18 tokens/s for the baseline, 33436.26 with a 0.2 GB buffer, 33608.93 with a 0.5 GB buffer, and 33656.72 with a 1.0 GB buffer. In particular, with only a 0.5 GB auxiliary buffer, the throughput degradation remains below 1\% relative to the baseline.
>
> ### 3. Coupling, Routing Flexibility, and Hard Tasks (W3, Q3)
> Thank you for this important concern. Our cross-layer coupling loss $R\_{cp}$  reduces harmful routing ambiguity rather than useful routing flexibility, which is especially helpful for complex reasoning because experts receive cleaner and more consistent token distributions. Empirically, this does not hurt flexible knowledge composition: on the reasoning-heavy benchmarks BBH, GPQA-Diamond, and GSM8K, adding $R_{sp}$ and $R\_{cp}$ yields consistent gains across backbones (Table 2). We also added a long-context benchmark RULER and again observed consistent improvements: 72.45 $\rightarrow$ 75.70 on DeepSeek-MoE, 78.35 $\rightarrow$ 80.95 on DeepSeek-V2-Lite, and 84.20 $\rightarrow$ 86.29 on Ling-mini-2.0.
> ### 4. What the Current Mechanistic Evidence Shows (W4, Q2)
> Thank you for this valuable suggestion. We have added a new visualization using code, math, and commonsense data to trace cross-layer expert pathways in the same MoE model. https://anonymous.4open.science/r/CDB4/  This suggests that the “coherent pathways” encouraged by $R\_{cp}$ are not merely arbitrary stable routes, but are more closely aligned with semantically differentiated processing patterns.
> ### 5. Risk of Early Routing Lock-In (Q4)
> Thank you for this important question. We have not observed evidence of harmful early lock-in within the tested coefficient range. A key reason is that $R\_{cp}$ is only a soft regularizer on routing probabilities, not a hard constraint on expert paths, and training is still dominated—especially in the early stage—by the main language modeling cross-entropy loss. Empirically, we observe gradual rather than premature sharpening: cross-layer coupling is already weakly present early in training and becomes progressively stronger over time (Figure 2).
> ### 6. Routing Diversity and Distributed Scalability (L1)
> Thank you for this important question. We respectfully believe it reflects a common misconception: the advantage of MoE is not maximal routing diversity itself, but effective expert specialization. If routing is overly diverse or unstable, similar tokens are scattered across many experts, so each expert receives a heterogeneous training distribution, learns mixed knowledge, and cannot specialize well under limited capacity. Our method reduces this harmful routing ambiguity rather than useful diversity. Proposition 4.1 further shows that this decorrelates expert updates. Proposition 5.1 and Theorems C.1/C.4 show that such coherence yields purer expert-specific training signals and strengthens specialization through depth. Empirically, this consistently improves perplexity and downstream performance while preserving load balancing (Tables 1/5, Tables 2/3, Figure 9).

---

> > ### Author Rebuttal · Reviewer_3n1a · 2026-04-02
> >
> > My concerns have been addressed.

---

> > > ### Author Response · Authors · 2026-04-02
> > >
> > > Dear Reviewer 3n1a,
> > >
> > > Thank you for your positive feedback and for taking the time to review our work. We truly appreciate your support and are glad that our clarifications addressed your concerns.
> > >
> > > Best regards,
> > >
> > > Authors of Submission 464

---

### Official Review · Reviewer_Tdyy · 2026-03-13

**Soundness:** 2
**Presentation:** 3
**Significance:** 2
**Originality:** 3
**Overall Recommendation:** 4
**Confidence:** 4

**Summary:**

This paper introduces two new plug-and-play loss functions to fix expert overlap and routing ambiguity in Mixture-of-Experts (MoE) models. The first is an intra-layer specialization loss. It stops experts in the same layer from learning redundant information by penalizing the cosine similarity of their activations. The second is a cross-layer coupling loss. It encourages the model to route tokens through the same expert paths across adjacent layers. The authors tested these losses on standard MoE and DeepSeek-style MoE architectures. Their results show improved perplexity and better accuracy on downstream tasks.

**Compliance With Llm Reviewing Policy:**

Affirmed.

**Final Justification:**

Author's rebuttal addressed my concerns. Therefore, I've raised my score to 4 weak accept.

**Key Questions For Authors:**

Q1. How does the cross-layer coupling loss handle adjacent layers that need to perform very different functions? Authors assume that tokens should take similar paths across adjacent layers. But what if layer 3 focuses on syntax and layer 4 focuses on logic? Forcing them to align might actually restrict the model's learning capacity. Did authors see any negative effects in specific layers?

Q2. Can authors provide more concrete data on the training overhead for large batch sizes? Authors state that the practical overhead is lightweight. However, computing pairwise cosine similarities for the intra-layer loss still takes extra memory during the forward pass. Did authors test this with very large batch sizes?

Q3. What happens if authors only apply these losses to specific layers? It is possible that the first few shallow layers do not need cross-layer coupling. Did authors try applying these losses to only the deep layers or only the shallow layers?

Q4. How does this method interact with other advanced routing strategies like Expert Choice Routing? Authors compared your method against standard load-balancing and a variance-based method. But Expert Choice Routing naturally helps balance loads and specialize experts too.

**Limitations:**

L1. The largest tested model has 7B parameters. A clear limitation is that the synergy between this paper's two losses might change, weaken, or cause instability at massive scales.

L2. The cross-layer coupling loss might not work well for MoE models that share experts across many layers, or models that use skip connections that bypass MoE layers entirely.

**Strengths And Weaknesses:**

Strengths:

- Using a cross-layer loss to explicitly build stable routing paths across depth is a clever and fresh idea.
- The authors support their empirical results with solid theoretical proofs, showing exactly how activation similarity aligns with gradient updates.
- The method is highly practical. It does not require changing the core model architecture or the router design, making it easy for others to adopt.

Weaknesses:

- The experiments mostly use small to medium models, maxing out at a 7B parameter model. It is not proven if these exact performance gains hold up for truly massive models.
- The cross-layer loss relies on the assumption that adjacent layers have similar token representations. If a specific model architecture forces large representation shifts between layers, this loss might fail or hurt performance.

---

> ### Author Rebuttal · Authors · 2026-03-31
>
> ### 1. Evidence Beyond the Current Scale (W1, L1)
>
> We pre-trained Qwen3-30B-A3B from scratch on 100B FineWeb-Edu tokens. Tables 2-3 in the paper show consistent fine-tuning gains at 16B and 30B.
>
> **Table 1: Qwen3-30B-A3B validation PPL.**
>
> |Loss|Valid PPL|
> |-|-|
> |$\mathcal{L}\_{lb}$|7.94|
> |$\mathcal{L}\_{lb,sp,cp}$|7.78|
> |$\mathcal{L}\_{lb,z}$|7.89|
> |$\mathcal{L}\_{lb,z,sp,cp}$|7.72|
>
> **Table 2: Qwen3-30B-A3B zero-shot accuracy.**
>
> |Loss|BoolQ|ARC-E|ARC-C|TruthfulQA-MC2|PIQA|MMLU|HellaSwag|Avg|
> |-|-|-|-|-|-|-|-|-|
> |$\mathcal{L}_{lb}$|67.43|69.93|41.72|46.87|72.24|32.27|57.24|55.39|
> |$\mathcal{L}_{lb,sp,cp}$|68.92|72.56|43.58|49.20|74.86|34.86|58.53|57.50|
> |$\mathcal{L}_{lb,z}$|68.06|71.84|42.56|46.63|73.92|33.28|56.91|56.17|
> |$\mathcal{L}_{lb,z,sp,cp}$|69.63|73.96|43.89|49.93|75.34|35.83|58.89|58.21|
>
> ### 2. Adjacent-Layer Continuity, Layer Functions, and Layer-Selective Application (W2, Q1, Q3)
>
> We clarify the design, present independent evidence, and report ablations.
>
> **(1)** $\mathcal{R}\_{cp}$ is a soft regularizer that stabilizes a few high-probability cross-layer transitions, not a hard alignment constraint. A token can invoke expert $i$ at layer $l$ and expert $j$ at layer $l{+}1$; $\mathcal{R}\_{cp}$ only asks that this transition pattern be consistent rather than noisy. The theoretical condition is "mild representation continuity," not "functional identity."
>
> **(2)** The reviewer's scenario ("layer 3 syntax, layer 4 logic") is an important boundary case, but current evidence suggests it may be uncommon in practice. In standard Transformers ($h^{l+1} = h^l + f(h^l)$), each block's contribution is small relative to the residual: adjacent layers show Procrustes similarity of 0.99 [4], later layers make only fine-grained adjustments rather than learning new computations [5], and representation similarity grows monotonically with proximity [6]. Since the router is linear in $h$, this smoothness carries over to routing decisions: pre-trained MoE models exhibit strong inter-layer expert affinity [7], Markovian routing dependencies [8], and local routing consistency across 20 MoE LLMs [9]. Middle layers also tolerate skipping or reordering [10], which would not hold if adjacent layers had truly orthogonal functions.
>
> **(3)** Both losses are plug-and-play and can be selectively applied to any subset of MoE layers. Applying $\mathcal{R}\_{cp}$ to shallow-only / deep-only / all layers gives PPLs of 12.42 / 12.32 / 12.27 (Vanilla, baseline 12.50). The gain is not tied to a specific layer range; broader application works best.
>
> $\mathcal{R}_{cp}$ is designed to reduce routing ambiguity and improve optimization stability; the evidence above suggests its underlying assumption holds across existing MoE architectures. Architectures with truly abrupt adjacent-layer shifts remain a boundary case; we will note this in the revision.
>
> ### 3. Practical Overhead at Large Batch Sizes (Q2)
>
> For the memory overhead, one can allocate a fixed-size buffer to compute cosine similarities for only one chunk of tokens at a time. After finishing the computation for that chunk, the buffer can be released and reused for the next chunk. In this way, the memory required for cosine-similarity computation is bounded by a controllable buffer size, rather than scaling with the total number of tokens in the batch. On Qwen3-30B-A3B (4 nodes / 32 H20 GPUs, EP$=32$, seq len $=4096$, batch $=2048$), throughput with a 0.5 GB buffer is 33609 tokens/s vs. 33868 baseline---below 1% degradation. We will include this in the revision.
>
> ### 4. Relationship to Expert Choice Routing (Q4)
>
> Expert Choice reverses routing direction (experts select tokens) and achieves perfect load balance, but load balance $\neq$ specialization---it has no mechanism to reduce co-activated expert overlap ($\mathcal{R}\_{sp}$) or cross-layer path consistency ($\mathcal{R}\_{cp}$). Our losses modify only the training objective and combine freely with Expert Choice. On medium-scale models with Expert Choice, adding $\mathcal{R}\_{sp}+\mathcal{R}\_{cp}$ reduces PPL from 12.72 to 12.39 (Vanilla) and 12.52 to 12.18 (DeepSeek-style).
>
> ### 5. Architectures with Nonstandard Token Paths (L2)
>
> **(1) Shared experts** do not affect our losses: both act only on routed experts; shared experts have no routing probability and lie outside scope. Our DeepSeek-style experiments confirm this. **(2) Token-level layer bypass** does not exist in current production MoE models (DeepSeek-V2/V3, Mixtral, Qwen-MoE, Llama 4, DBRX, Jamba all route every token through every layer). Research architectures with bypass (MoD [11]) have not been deployed. If adopted, $\mathcal{R}\_{cp}$ would need adaptation; we will note this as a limitation.
>
> **References:** [4] arXiv:2405.12250 [5] arXiv:2505.13898 [6] arXiv:2406.14479 [7] arXiv:2401.08383 [8] arXiv:2503.04398 [9] arXiv:2505.16056 [10] arXiv:2407.09298 [11] arXiv:2404.02258

---

> > ### Author Rebuttal · Reviewer_Tdyy · 2026-04-03
> >
> > My concerns have been adequately addressed.

---

> > > ### Author Response · Authors · 2026-04-04
> > >
> > > Dear Reviewer Tdyy,
> > >
> > > Thank you for your careful review and positive feedback. We are very glad that our response has fully clarified your concerns, and we would kindly ask you to take this into account when adjusting your final justification accordingly.
> > >
> > > Best regards,
> > >
> > > Authors of Submission 464

---

### Official Review · Reviewer_QRbZ · 2026-03-13

**Soundness:** 3
**Presentation:** 3
**Significance:** 3
**Originality:** 2
**Overall Recommendation:** 4
**Confidence:** 4

**Summary:**

This work proposes two plug-and-play regularization losses, an intra-layer specialization loss ($R_{sp}$) and a cross-layer coupling loss ($R_{cp}$), to improve expert specialization and routing efficiency in sparse Mixture-of-Experts (MoE) models without modifying router or model architectures. $R_{sp}$ penalizes cosine similarity between experts' SwiGLU activations on the same token, pushing experts to develop complementary and functionally distinct representations. $R_{cp}$ maximizes joint top-k routing probabilities across adjacent layers, encouraging tokens to follow consistent expert pathways. Both losses are architecture-agnostic and orthogonal to standard load-balancing objectives. Experiments across pre-training, LoRA SFT, and full-parameter fine-tuning, as well as zero-shot benchmarks, consistently demonstrate perplexity reductions and downstream task gains.

**Compliance With Llm Reviewing Policy:**

Affirmed.

**Final Justification:**

My main concerns have been addressed. I therefore retain my recommendation of acceptance.

**Key Questions For Authors:**

Please see the weaknesses above.

**Limitations:**

Yes.

**Strengths And Weaknesses:**

**Strengths**

1. The paper is well-structured and clearly written, making the motivation, methodology, and theoretical contributions easy to follow.

2. The proposed method is well-motivated by identifying two concrete failure modes in MoE training — expert overlap (redundant activations across co-activated experts) and routing ambiguity (inconsistent token-to-expert assignments).

3. The paper provides theoretical grounding, including Proposition 4.1 linking activation similarity to gradient alignment, a closed-loop feedback mechanism (Theorems C.1 and C.4) formalizing the mutual reinforcement between specialization and routing decisiveness, and explicit compatibility proofs with standard load-balancing objectives.

4. Both losses are simple and lightweight. $R_{sp}$ reuses cached activations with no extra matrix multiplications, and $R_{cp}$  operates only on scalar routing scores. The computational complexity O(k²d) is discussed and empirically validated in Appendix H.

5. The paper presents comprehensive experiments and ablation studies across pre-training at three model scales, LoRA SFT on three independent 16B-class backbones, full-parameter fine-tuning on a 30B model, and zero-shot evaluation across multiple benchmarks.


**Weaknesses**
1. Proposition 5.1 and Theorems C.1/C.4 are derived under the top-1 routing setting (k=1), while all main experiments use k=2 or k=4. The theoretical gap between the assumed and actual routing settings should be discussed.

2.  The cross-layer coupling loss $R_{cp}$ only enforces routing consistency between immediately adjacent layers (l, l+1), with no mechanism to ensure that the coupled expert pairs are semantically meaningful — two experts can achieve high joint probability through correlated but redundant routing without genuinely improving functional specialization.

3. The zero-shot benchmark improvements reported in Table 6 are marginal in absolute terms (average gains of ~0.007–0.009). It is difficult to determine whether the observed gains are reliably attributable to the proposed losses.

---

> ### Author Rebuttal · Authors · 2026-03-31
>
> ### 1. Bridging the Top-1 Theory and Top-k Practice (W1)
> The top-1 setting was chosen as the simplest case for exposing the specialization--routing feedback loop. We believe the gap to top-k is narrower than it may appear, and we clarify this from three angles.
>
> **The formal results are not locked to $k{=}1$.** Proposition 5.1 already uses general activated expert sets $\mathbb{A}\_i^{(l)}$ with no constraint on $|\mathbb{A}\_i^{(l)}|$, so it applies to top-k as written. The per-expert gradient $\frac{\partial L}{\partial z\_e} = g\_e(\ell\_e - L)$ derived in the proof of Theorem C.1 holds for each expert inside the top-k softmax---the direction that pushes probability mass toward lower-loss experts does not change with $k$. Theorem C.4 uses single-expert assignments $C\_\ell(t)$; in top-k this becomes subset selection $S\_\ell(t) \in \binom{[E]}{k}$, and the bound linking later-layer routing quality to earlier-layer error still applies, though a fully rigorous version would need the coupling coefficient redefined over subset matchings. More broadly, a natural way to see the extension is to treat each active $k$-subset as a single "virtual expert": $E$ physical experts under top-k induce $\binom{E}{k}$ possible active subsets, and the top-1 analysis applies to these composite units. We will add a proof sketch along these lines in revision.
>
> This combinatorial view has precedent. DeepSeekMoE [1] motivates fine-grained segmentation via $\binom{N}{k}$; Nguyen et al. [2] partition the input space into $\binom{k_*}{K}$ regions, each with a fixed active subset; Yang et al. [3] rewrite MoE layers as structured-sparse MLPs whose sparsity pattern is determined by the top-k choice. All three treat top-k as subset selection, not a fundamentally different regime.
>
> Top-1 experiments close the empirical gap. On a 1B MoE model with top-1 routing, Vanilla MoE validation PPL improves from 13.41 (baseline) to 13.18 / 13.29 / 13.09 with $\mathcal{R}_{cp}$ / $\mathcal{R}\_{z}$ / $\mathcal{R}\_{z,cp}$; DeepSeek-style MoE improves from 13.17 to 12.99 / 13.04 / 12.87. The same pattern holds at top-k in the main experiments, confirming that the theoretical predictions are not an artifact of the $k{=}1$ assumption.
>
> ### 2. What the Cross-Layer Coupling Actually Encourages (W2)
> $\mathcal{R}\_{cp}$ is designed to promote optimization-relevant path consistency, not semantic alignment between expert pairs. It does not dictate what experts should learn; semantic learning is still driven by the main cross-entropy objective. Its role, formalized by Proposition 5.1, is to stabilize cross-layer routing transitions so that specialization formed within each layer can propagate across depth. This is also why $\mathcal{R}_{sp}$ matters here: by reducing activation overlap among co-activated experts, it discourages correlated-but-redundant routing and makes coherent pathways more likely to reflect functionally differentiated experts. We also do a new visualization (see 3n1a W4)
> ### 3. How Reliable Are the Downstream Gains? (W3)
> We acknowledge that the absolute zero-shot gains are modest. However, we note the following evidence of reliability:
> - Results are averaged over 7 benchmarks and 3 random seeds per setting;
> - Gains appear across both Vanilla and DeepSeek-style MoE architectures;
> - The hyperparameter sensitivity analysis (Appendix E.4) shows that gains are stable over a wide range of $\lambda_{sp}$ and $\lambda\_{cp}$ values (PPL varies by <1% over a 10$\times$ range of $\lambda\_{cp}$).
> The scalability experiments (Figures 4-5) offer a different angle: with our losses, activating only $N{=}6$ experts already outperforms the baseline with $N{=}10$ (PPL 12.13 vs. 12.15), and a smaller expert pool of $E{=}96$ surpasses the baseline with $E{=}112$ (PPL 11.93 vs. 11.94). This shows improved expert utilization---fewer experts needed for the same or better performance-which directly reduces deployment cost and inference latency.
> Another factor that may limit the improvement on zero-shot tasks is the relatively small model size (1B parameters) and insufficient training data(30B tokens from the C4-English dataset). To further investigate this issue, we pretrained Qwen3-30B-A3B from scratch using 100B tokens from the FineWeb-Edu dataset. The zero-shot evaluation results are reported  in Table 1.
> Table 1: Zero-shot accuracy of  Qwen3-30B-A3B pretrained from scratch.
> |Loss|BoolQ|ARC-E|ARC-C|TruthfulQA-MC2 | PIQA | MMLU |HellaSwag|Avg|
> |-|-|-|-|-|-|-|-|-|
> |$\mathcal{L}_{lb}$|67.43|69.93|41.72|46.87|72.24|32.27|57.24|55.39|
> |$\mathcal{L}_{lb,sp,cp}$| 68.92|72.56|43.58|49.20|74.86|34.86|58.53|57.50|
> |$\mathcal{L}_{lb,z}$|68.06|71.84|42.56|46.63|73.92|33.28|56.91|56.17|
> |$\mathcal{L}_{lb,z,sp,cp}$|69.63|73.96|43.89|49.93|75.34|35.83|58.89|58.21|
>
> Pretraining the MoE model with our losses improves the average downstream task performance by 2.11 and 2.04 points, respectively.
>
> **References:** [1] arXiv:2401.06066 [2] arXiv:2309.13850 [3] arXiv:2503.07639

---

> > ### Author Rebuttal · Reviewer_QRbZ · 2026-04-04
> >
> > My concerns have been addressed; the manuscript should be revised accordingly

---

> > > ### Author Response · Authors · 2026-04-04
> > >
> > > Dear Reviewer QRbZ,
> > >
> > > Thank you for your careful review and for your positive feedback. We are very glad that our response has fully clarified your concerns.
> > >
> > > Best regards,
> > >
> > > Authors of Submission 464

---

### Decision · Program_Chairs · 2026-04-30

**Decision:**

Accept (regular)

**Comment:**

The manuscript was evaluated by four reviewers, all of whom reached a positive overall assessment after considering the rebuttal.

The reviewers found the work to be well motivated, theoretically sound, and supported by promising empirical results.

The Area Chair concurs with this consensus and recommends acceptance.

The authors are nonetheless encouraged to carefully incorporate the reviewers’ suggestions in the final camera-ready version.

Congrats!